# Cross-national analyses require additional controls to account for the non-independence of nations

Scott Claessens [1] ✉, Thanos Kyritsis [1] & Quentin D. Atkinson [1,2] ✉

Cross-national analyses test hypotheses about the drivers of variation in national outcomes. However, since nations are connected in various ways, such as via spatial proximity and shared cultural ancestry, cross-national analyses often violate assumptions of non-independence, inflating false positive rates. Here, we show that, despite being recognised as an important statistical pitfall for over 200 years, cross-national research in economics and psychology still does not sufficiently account for non-independence. In a review of the 100 highest-cited cross-national studies of economic development and values, we find that controls for non-independence are rare. When studies do control for non-independence, our simulations suggest that most commonly used methods are insufficient for reducing false positives in non-independent data. In reanalyses of twelve previous cross-national correlations, half of the estimates are compatible with no association after controlling for non-independence using global proximity matrices. We urge social scientists to sufficiently control for non-independence in cross-national research.

Nations are an important population unit structuring modern human life. The nation in which someone is born has a large effect on what they can expect out of life, including their income level[1], life expectancy[2], mental health[3], subjective well-being[4], and educational attainment[5]. Nations are also among the most important determinants of human cultural variation, with greater cultural similarity within than outside national borders[6].

Given the importance of nations for structuring human behaviour, there is justifiably huge interest in statistical analyses that attempt to predict variation in national outcomes of all kinds. At the time of writing, a search on the Web of Science for the term "cross-national" in titles or abstracts returned over 13,000 unique hits. The standard practice for cross-national analyses is to conduct bivariate correlations or multiple regressions with individual data points representing different nations. Such analyses widen the scope of social science beyond Western populations[7,8] and have been used to study, among other topics, the causes of variation in the economic wealth of nations[9–12], global patternings of cultural norms and values[13–16], and the

universality and diversity of human behaviour and psychology around the world[17–20].

However, cross-national analyses are complicated by the fact that nations are not statistically independent data points. Unlike independent random samples from a population, nations are related to one another in a number of ways. First, nations that are closer to one another tend to be more similar than distant nations. This phenomenon is known as spatial non-independence[21], and it occurs because nations in close spatial proximity share characteristics due to local cultural diffusion of ideas[22] and regional variation in climate and environment[21]. For example, the neighbouring African nations Zambia and Tanzania have similar levels of terrain ruggedness, which has been used to partially explain their similar levels of economic development[23]. This pattern conforms to Tobler's first law of geography: "everything is related to everything else, but near things are more related than distant things"[24] (p. 236).

Second, nations with shared cultural ancestry tend to be more similar than culturally unrelated nations. This is known as cultural

[1]School of Psychology, University of Auckland, Auckland, New Zealand. [2]School of Anthropology and Museum Ethnography, University of Oxford, Oxford, United Kingdom. ✉e-mail: scott.claessens@gmail.com; q.atkinson@auckland.ac.nz

phylogenetic non-independence[25–27] and occurs because related nations share cultural traits inherited via descent from a common ancestor. Shared cultural ancestry can result in a form of pseudoreplication, whereby multiple instances of the same trait across nations are merely duplicates of the ancestral original[25,28]. For example, the related island nations Tonga and Tuvalu share similar languages and customs due to cultural inheritance from a common Polynesian population dating back more than 1000 years[29,30]. More recently, shared ancestry explains similarities in institutions, norms, technologies, and values between colonial settlements and their colonisers (e.g., English, French, Spanish, and Portuguese settlements of the Americas). Importantly, these deep cultural connections between nations often span large geographic distances around the world. Tonga and Tuvalu share cultural traits despite being separated by over 1500 kilometres of ocean, and South American and European nations remain culturally similar today despite their locations on two separate continents. Shared cultural ancestry must therefore be considered independently of spatial proximity in the study of nations.

Spatial and cultural phylogenetic non-independence between nations make cross-national inference challenging. A fundamental assumption of regression analysis is that model residuals should be independently and identically distributed[31]. But without accounting for spatial or cultural non-independence between nations, model residuals can show structure that remains unaccounted for, violating this assumption. Treating nations as independent can thus inflate false positive rates[32], producing spurious relationships between variables that, in fact, only indirectly covary due to spatial or cultural connections[33] (see Supplementary Fig. 1 for an example causal model).

Non-independence is widely acknowledged in fields that routinely deal with spatially or culturally structured data. In ecology and sociology, studies with regional-level data use a variety of autoregressive models designed to account for spatial patternings[34,35]. In anthropology, researchers have recognised cultural non-independence as an important statistical pitfall for over 200 years, with issues of cultural pseudoreplication being identified in early comparative studies of marriage practices across societies[25]. In the twentieth century, anthropologists began to emphasise that human societies do not develop independently but rather exist in a globally interconnected world system linked by shared history and cultural ancestry[36]. In order to minimise the confounding effects of this non-independence in comparisons of human societies, researchers compiled the Standard Cross-Cultural Sample of 186 cultures, which deliberately avoids sampling closely related cultures[28], though it is difficult to completely remove spatial and cultural dependencies[37,38] and the smaller sample of cultures reduces statistical power. Today, anthropologists borrow phylogenetic comparative methods from evolutionary biology, such as phylogenetic least squares regression[39], when comparing societies, treating culturally related societies in the same way as biologists treat genetically related species[40,41].

At the national level, recent reanalyses have revealed that several cross-national relationships reported in economics and psychology do not hold when controlling for non-independence between nations. One working paper replicated 25 analyses of persistence in economics, in which modern national outcomes are regressed against historical characteristics of those nations, and found that over half of the relationships were attenuated when controlling for spatial non-independence[42]. Another replication study found that many of the widely publicised relationships between national-level pathogen prevalence and political institutions and attitudes fail to hold when controlling for various kinds of non-independence[43]. These reanalyses, and others[44–46], raise the question: how widespread a concern is non-independence in studies of national-level outcomes?

To address this question, we consider national-level variables of general interest across the social sciences: economic development and cultural values. These variables are frequently included as both outcomes and predictors in cross-national studies in economics and psychology[9–16]. First, we demonstrate that economic development and cultural values are spatially and culturally non-independent across nations, emphasising the need to control for non-independence. Second, we review the 100 highest-cited cross-national studies of economic development and cultural values and determine baseline rates of controlling for non-independence in the literature. Third, we run simulations to determine whether common methods of dealing with non-independence in the literature sufficiently reduce false positive rates. Fourth, we reanalyse 12 previous cross-national analyses of economic development and cultural values from our literature review, incorporating global geographic and linguistic proximity matrices as controls for spatial and cultural non-independence.

## Results

### National-level economic development and cultural values are spatially and culturally non-independent

In order to motivate our research question, it is important to first quantify the degree of spatial and cultural non-independence for economic development and cultural values around the world. If these variables are independent or only weakly non-independent, then the issue might be safe to ignore. However, if they are more strongly non-independent, then there is a possibility that non-independence could be confounding cross-national inferences.

To this end, we used Bayesian multilevel models to simultaneously estimate geographic and cultural phylogenetic signals for a range of economic development and cultural values variables. For economic development, we focused on the Human Development Index[47], gross domestic product per capita, annual gross domestic product per capita growth, and the Gini index of income inequality. For cultural values, we focused on two primary dimensions of cultural values from the World Values Survey, traditional vs. secular values and survival vs. self-expression values[16], as well as cultural tightness[14] and individualism[15]. These variables are not intended to be a comprehensive list of all national-level variables included in cross-national research but rather an illustrative set of variables that are widely used in the literature.

For all of these variables, we found that a substantial proportion of national-level variation was explained by spatial proximity and/or shared cultural ancestry between nations (Fig. 1; see Supplementary Table 1 for numerical results). Signal estimates were often strong, with spatial proximity and shared cultural ancestry frequently explaining over half of the national-level variation. For spatial proximity, Bayes Factors indicated strong evidence that the geographic signal estimates differed from zero for all economic development variables and traditional values. However, the evidence was only equivocal for survival values and individualism, and strong evidence was found that the geographic signal estimate for tightness was equal to zero. For shared cultural ancestry, Bayes Factors indicated strong evidence that the cultural phylogenetic signal estimates differed from zero for all economic development and cultural values variables except for gross domestic product per capita growth, for which the evidence was equivocal. These findings emphasise the need to account for spatial and cultural phylogenetic non-independence in cross-national analyses of economic development and cultural values.

### Previous cross-national analyses have not sufficiently accounted for non-independence

Given that economic development and cultural values show evidence of geographic and cultural phylogenetic signals, have cross-national analyses sufficiently accounted for this non-independence? To assess this, we searched the published literature for articles that combined the search terms "economic development" or "values" with the search terms "cross-national", "cross-cultural", or "cross-country". We removed articles that did not report original research, were not

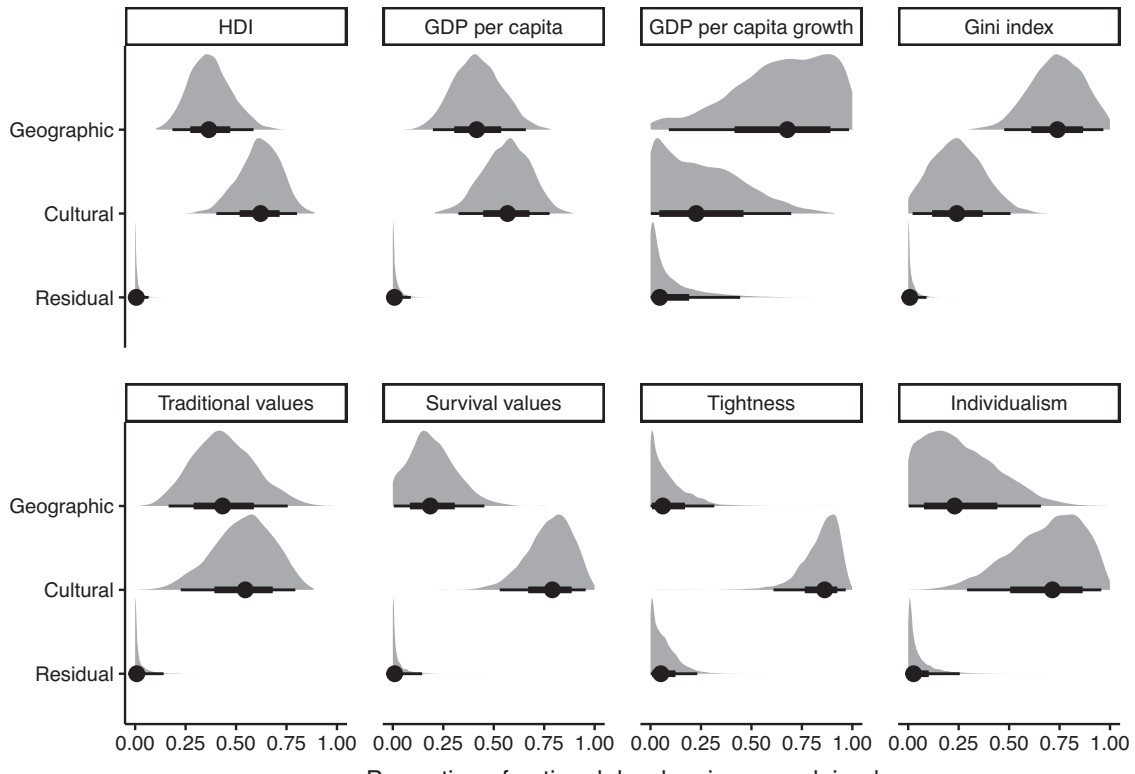

**Fig. 1 | Posterior estimates of geographic and cultural phylogenetic signal for a range of economic development and cultural values variables.** Geographic and cultural phylogenetic signals are operationalised as the proportion of national-level variance explained by geographic and linguistic proximity matrices. Grey ridges are full posterior distributions, points are posterior median values, and black lines are equal-tailed 50% and 95% credible intervals. Number of observations from these models are as follows: HDI ($n = 1449$), GDP per capita ($n = 10289$), GDP per capita growth ($n = 9755$), Gini index ($n = 1826$), traditional values ($n = 277676$), survival values ($n = 277676$), tightness ($n = 57$), and individualism ($n = 67$). HDI Human Development Index, GDP gross domestic product.

relevant to economic development or cultural values, or did not report at least one cross-national analysis. We then retained the 100 articles (50 for economic development, 50 for cultural values) with the highest annual rate of citations (see Supplementary Data 1). For each of these highly cited articles, we exhaustively recorded every cross-national analysis reported in the main text ($n = 4308$), identifying in each case whether or not the analysis attempted to control for spatial, cultural, or any other form of non-independence between nations (see "Methods" for detailed search criteria and coding decisions).

The results of our literature review show that most published articles containing cross-national analyses make no attempt to account for statistical non-independence. Figure 2a plots the proportion of articles that contain at least one cross-national analysis accounting for non-independence. We find that 42% of economic development articles contain at least one attempt to control for non-independence (two-tailed 95% bootstrap confidence interval [0.30 0.54]), while this proportion decreases to only 8% for cultural values articles (two-tailed 95% bCI [0.02 0.16]). Both kinds of articles are most likely to use regional fixed effects (e.g., continent fixed effects) to account for non-independence, but some articles also include controls for spatial distance (e.g., latitude) and shared cultural history (e.g., colony status). These proportions are even lower when focusing on the full sample of 4308 analyses: only 5% (equal-tailed 95% credible interval [0.02 0.13]) of individual economic development analyses and 1% (equal-tailed 95% CI [0.00 0.02]) of individual cultural values analyses are estimated to control for non-independence (Supplementary Fig. 2).

While our review contains articles from journals with a range of impact factors, our estimates could be biased downwards by analyses published in lower-impact outlets with more relaxed standards for issues like non-independence. It is also possible that, since our

literature review goes back as far as 1993, our estimates are being biased downwards by earlier studies and that controls for non-independence have increased over time with methodological advancements and greater awareness of the issue. To test these possible explanations for our low estimates, we fitted logistic regression models to the data from the review, including log journal impact factor and publication year as separate predictors. Interestingly, we found that, if anything, studies from higher-impact journals were less likely to include at least one control for non-independence than studies from lower-impact journals, both for studies of economic development (b = −0.38, equal-tailed 95% CI [−0.87 0.09]) and for studies of cultural values (b = −0.56, equal-tailed 95% CI [−1.10 −0.05]; Fig. 2b). Moreover, splines revealed no relationship between publication year and the probability of including at least one control for non-independence, both for studies of economic development (b = −0.13, equal-tailed 95% CI [−1.10 0.82]) and for studies of cultural values (b = −0.02, equal-tailed 95% CI [−1.00 0.96]; Fig. 2c).

## Common methods of controlling for non-independence are insufficient for reducing false positive rates in non-independent data

Our literature review revealed that most cross-national analyses in the literature do not control for spatial or cultural phylogenetic non-independence. When they do, they tend to include controls like latitude and regional fixed effects. Do these methods sufficiently account for statistical non-independence?

To compare the efficacy of different methods in the literature, we conducted a simulation study. We simulated national-level datasets ($n = 236$ nations) with varying degrees of spatial or cultural phylogenetic autocorrelation (i.e., non-independence) for outcome and

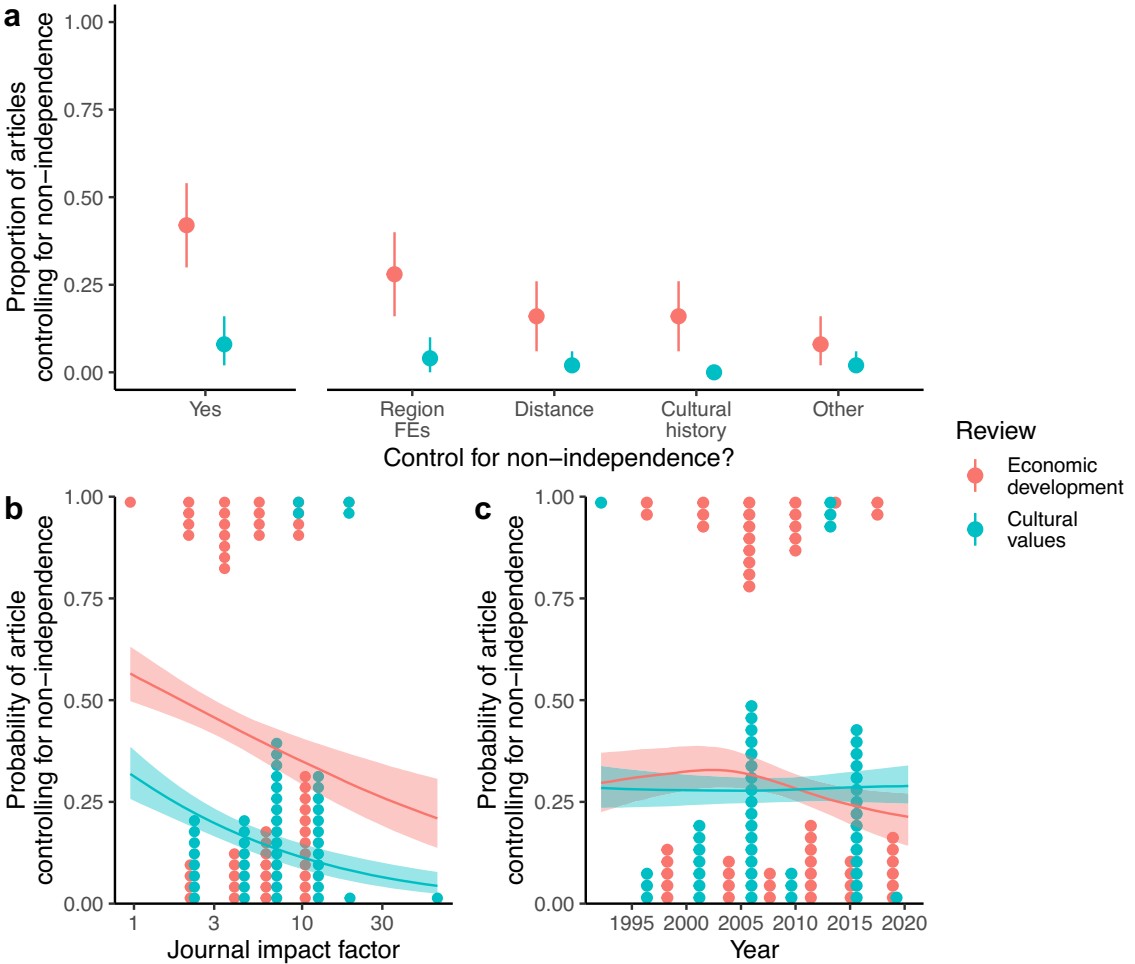

**Fig. 2 | Results from literature review of 100 highly cited cross-national studies of economic development (red) and cultural values (blue). a** Proportion of articles containing at least one analysis accounting for non-independence, overall and split by common methods of controlling for non-independence. Points represent raw proportions of articles, and ranges represent two-tailed 95% bootstrap confidence intervals (*n* = 1000 bootstrap samples). **b** The association between journal impact factor and the probability that an article contains at least one analysis accounting for non-independence. **c** Estimated trend over time for the probability that an article contains at least one analysis accounting for non-independence. Lines and shaded areas are posterior median regression lines and equal-tailed 50% credible intervals from Bayesian logistic regression models (*n* = 100 observations). Dots represent raw counts of individual articles that did (top) or did not (bottom) account for non-independence. Region FEs region fixed effects.

predictor variables but with no direct causal relationship between the variables. We then fitted naive regressions without controls to these datasets, as well as regression models with controls for latitude, longitude, and continent fixed effects. Despite not being identified in our literature review, we also included other methods that are often used in the literature to account for non-independence. Additional spatial controls included the mean of the predictor variable within a surrounding 2000-km radius[48] and Conley standard errors[49,50] based on geographic distances between nations[19,48]. Additional cultural controls included fixed effects for the language families of the majority-spoken languages in each nation[51] and Conley standard errors based on genetic distances between nations[19,48]. These fixed effect approaches attempt to account for non-independence by holding geographic location constant (latitude, longitude), discarding between-region variation and exploiting only local variation (continent fixed effects, mean of surrounding 2000 km), or correcting standard errors for autocorrelation post hoc while leaving model coefficients unchanged (Conley standard errors).

Beyond fixed effect approaches, we also fitted Bayesian random effects regressions that explicitly model spatial and/or cultural phylogenetic non-independence by allowing nations to covary according to geographic and/or linguistic proximity matrices. The geographic

proximity between nations is calculated from inverse distances between longitude and latitude coordinates. Linguistic proximity between nations is calculated from a global phylogenetic tree that represents hierarchical relationships of genealogical descent for all languages in the world. For each pair of nations, we calculated inverse phylogenetic distances (i.e., number of branches separating two taxa) between all languages spoken in that nation pair and produced an average linguistic proximity score weighted by the percentages of speakers within those nations. To include the resulting geographic and linguistic proximity matrices in our models, we included a Gaussian process[52,53] over latitude and longitude values and/or assumed that nation random intercepts were correlated in proportion to their linguistic proximity[54]. These random effects approaches attempt to account for non-independence by modelling the covariance between nations that is induced by their geographic or linguistic connections.

Figures 3 and 4 plot the estimated false positive rates from our simulation study, split by different methods and different degrees of spatial or cultural phylogenetic autocorrelation (see Supplementary Tables 2 and 3 for numerical results and Supplementary Figs. 3 and 4 for full distributions of effect sizes under strong autocorrelation). Across all model types, false positive rates were measured as the proportion of models that estimated a slope with a two-tailed 95%

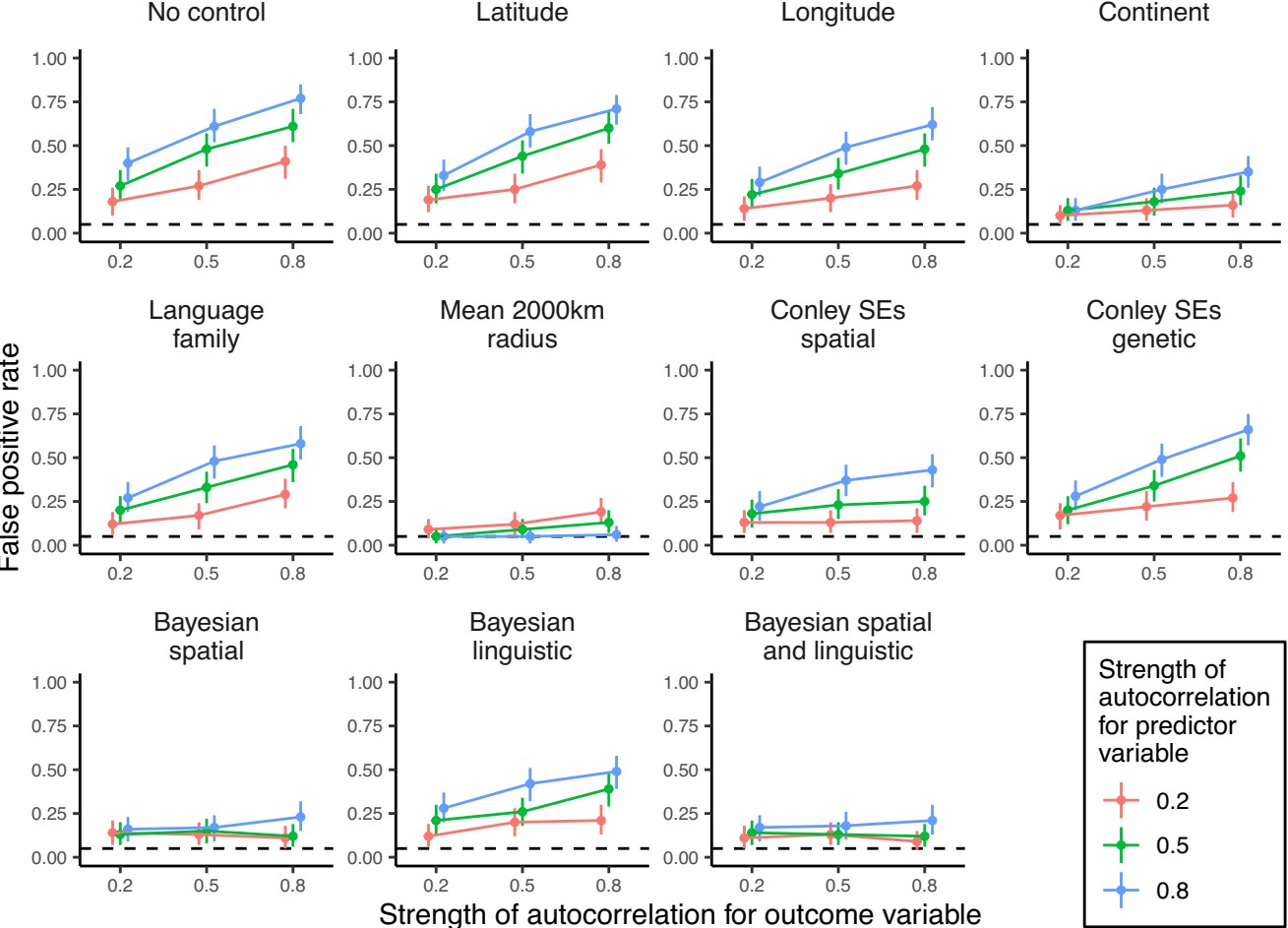

**Fig. 3 | False positive rates for different methods of controlling for spatial non-independence in our simulation study.** For simulated outcome and predictor variables, we systematically varied the strength of spatial autocorrelation from weak (0.2) to moderate (0.5) to strong (0.8). We simulated 100 datasets per parameter combination and fitted different models to each dataset. False positive rates were operationalised as the proportion of models that estimated a slope with a two-tailed 95% confidence/credible interval excluding zero. Points represent raw proportions of false-positive models, ranges represent two-tailed 95% bootstrap confidence intervals (*n* = 1000 bootstrap samples), and dashed lines indicate the 5% false positive rate that is expected due to chance. Colours indicate whether the strength of autocorrelation for the predictor variable is 0.2 (red), 0.5 (green) or 0.8 (blue). SEs standard errors.

confidence/credible interval excluding zero, falsely inferring a relationship when none is present. For reference, weak autocorrelation in our simulation is comparable to the geographic signal for survival values in Fig. 1 (i.e., 20% of the national-level variance is explained by non-independence), while moderate and strong levels of autocorrelation are comparable to the cultural phylogenetic signal for traditional and survival values, respectively (i.e., 50% and 80% of the national-level variance is explained by non-independence).

Our simulation study revealed that with at least moderate degrees of spatial or cultural phylogenetic autocorrelation for both outcome and predictor variables, naive regression models produce false positive rates above chance levels. This false positive rate increases as the degree of autocorrelation increases. With strong spatial autocorrelation for both outcomes and predictors, false positive rates reach as high as 77%. We find a slightly lower false positive rate under strong cultural phylogenetic autocorrelation, though this false positive rate is still greater than expected by chance (36%).

Most methods common in the literature do not reduce these high false positive rates. With strong spatial autocorrelation for both outcome and predictor variables, false positive rates remain above 50% when controlling for latitude, longitude, and language family fixed

effects (Fig. 3). Similarly, Conley standard error corrections based on spatial and genetic distances do not reduce false positive rates below 40% under strong spatial autocorrelation. The most effective fixed effects methods are continent fixed effects, which continue to produce a false positive rate of 35% under strong spatial autocorrelation, and controlling for the mean of the predictor variable within a 2000-km radius, which eliminates false positives under even strong spatial autocorrelation (6%). However, additional simulations revealed that these reductions in false positive rates come at the cost of lower statistical power (see Supplementary Figs. 5–7). In additional simulations where the true relationship between the predictor and outcome variable was known, power analyses showed that both continent fixed effects and the 2000-km radius control had less than 80% power to detect moderate true correlations (*r* = 0.3) under strong spatial autocorrelation.

By contrast, Bayesian spatial Gaussian process regression with longitude and latitude strikes a balance between reducing false positives and retaining high statistical power to detect true effects. This approach reduces false positives to 15% under moderate spatial autocorrelation and 23% under strong spatial autocorrelation. Random effects models that additionally account for linguistic proximity

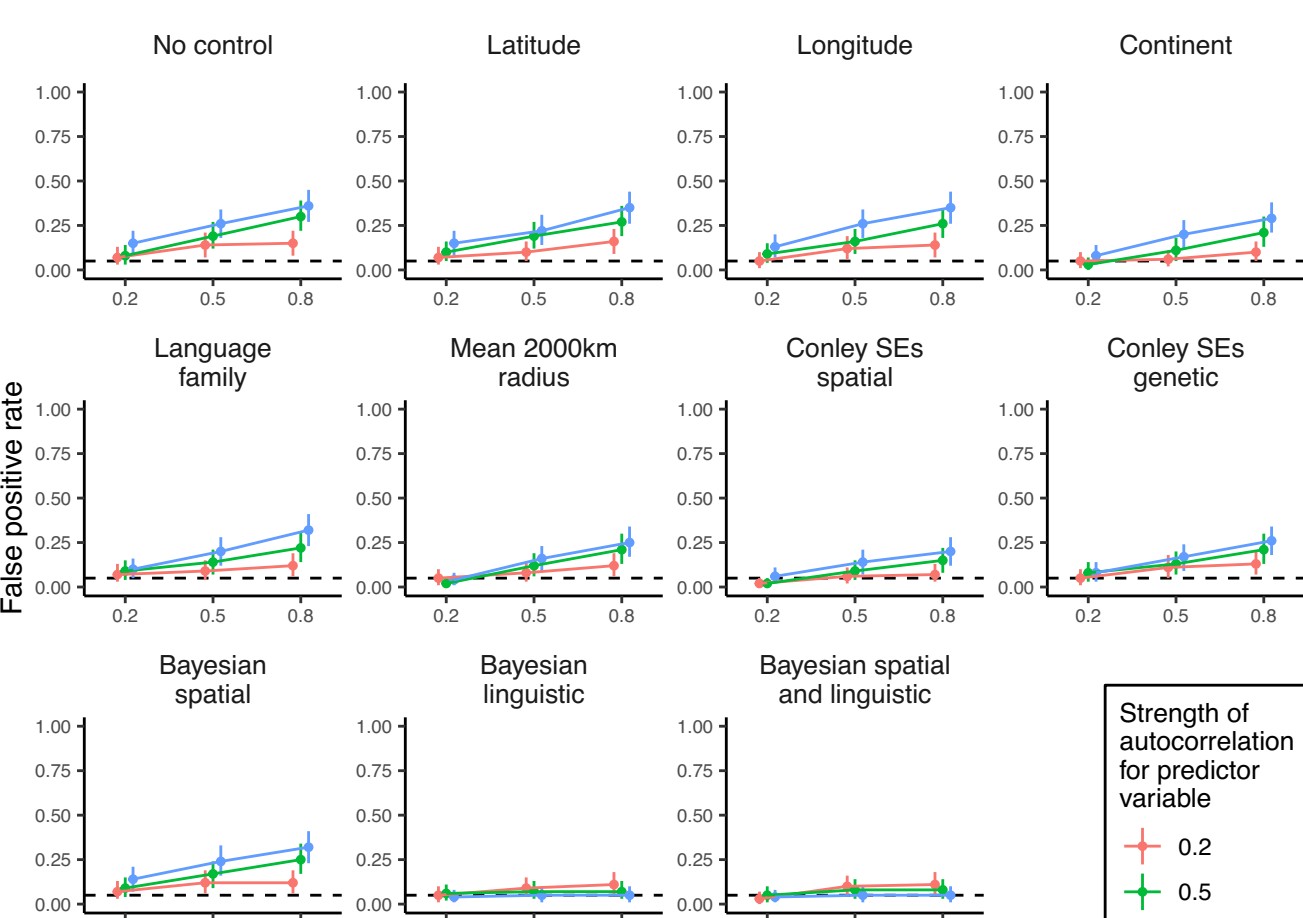

**Fig. 4 | False positive rates for different methods of controlling for cultural phylogenetic non-independence in our simulation study.** For simulated outcome and predictor variables, we systematically varied the strength of cultural phylogenetic autocorrelation from weak (0.2) to moderate (0.5) to strong (0.8). We simulated 100 datasets per parameter combination and fitted different models to each dataset. False positive rates were operationalised as the proportion of models that estimated a slope with a two-tailed 95% confidence/credible interval excluding zero. Points represent raw proportions of false-positive models, ranges represent two-tailed 95% bootstrap confidence intervals ($n = 1000$ bootstrap samples), and dashed lines indicate the 5% false positive rate that is expected due to chance. Colours indicate whether the strength of autocorrelation for the predictor variable is 0.2 (red), 0.5 (green) or 0.8 (blue). SEs standard errors.

between nations perform equally well, though models with only linguistic covariance continue to produce false positives. False positives are not completely eliminated with these random effects models. Nevertheless, these methods have at least 80% power to detect moderate ($r = 0.3$) and large ($r = 0.5$) true correlations between variables under all levels of spatial autocorrelation (see Supplementary Figs. 5–7).

In our simulation of cultural phylogenetic non-independence, we find that none of the fixed effects methods reduce false positive rates (Fig. 4). Controls for latitude, longitude, continent fixed effects, the mean of the predictor variable in a 2000-km radius, and Conley standard error corrections based on spatial and genetic distances do little to change false positive rates. Even language family fixed effects continue to produce a false positive rate of 32% under strong cultural phylogenetic autocorrelation. By contrast, models with random effects covarying according to linguistic proximity completely eliminate false positives across all degrees of cultural phylogenetic autocorrelation. This approach is also the only method that is able to detect large true correlations ($r = 0.5$) with at least 80% power (see Supplementary Figs. 8–10). Random effects models that additionally account for geographic proximity between nations perform equally well, though

models with only a spatial Gaussian process continue to produce false positives.

## Key findings in the literature are not robust to reanalysis with more rigorous methods

Our literature review and simulation study have shown that controls for non-independence are rare in cross-national studies of economic development and cultural values, and when studies do attempt to control for non-independence, the methods typically used are unable to sufficiently reduce false positive rates. This raises the worrying possibility that the cross-national literature in economics and psychology is populated with spurious relationships.

To determine how widespread this issue of spurious cross-national relationships might be, we reanalysed a subset of 12 previous cross-national analyses from our literature review, controlling for spatial and cultural phylogenetic non-independence using global geographic and linguistic proximity matrices. Out of the 100 papers included in our literature review, primary or secondary data were publicly available for 47 papers. We attempted to replicate key statistically significant cross-national correlations from these papers—mostly initial bivariate regression specifications without covariates

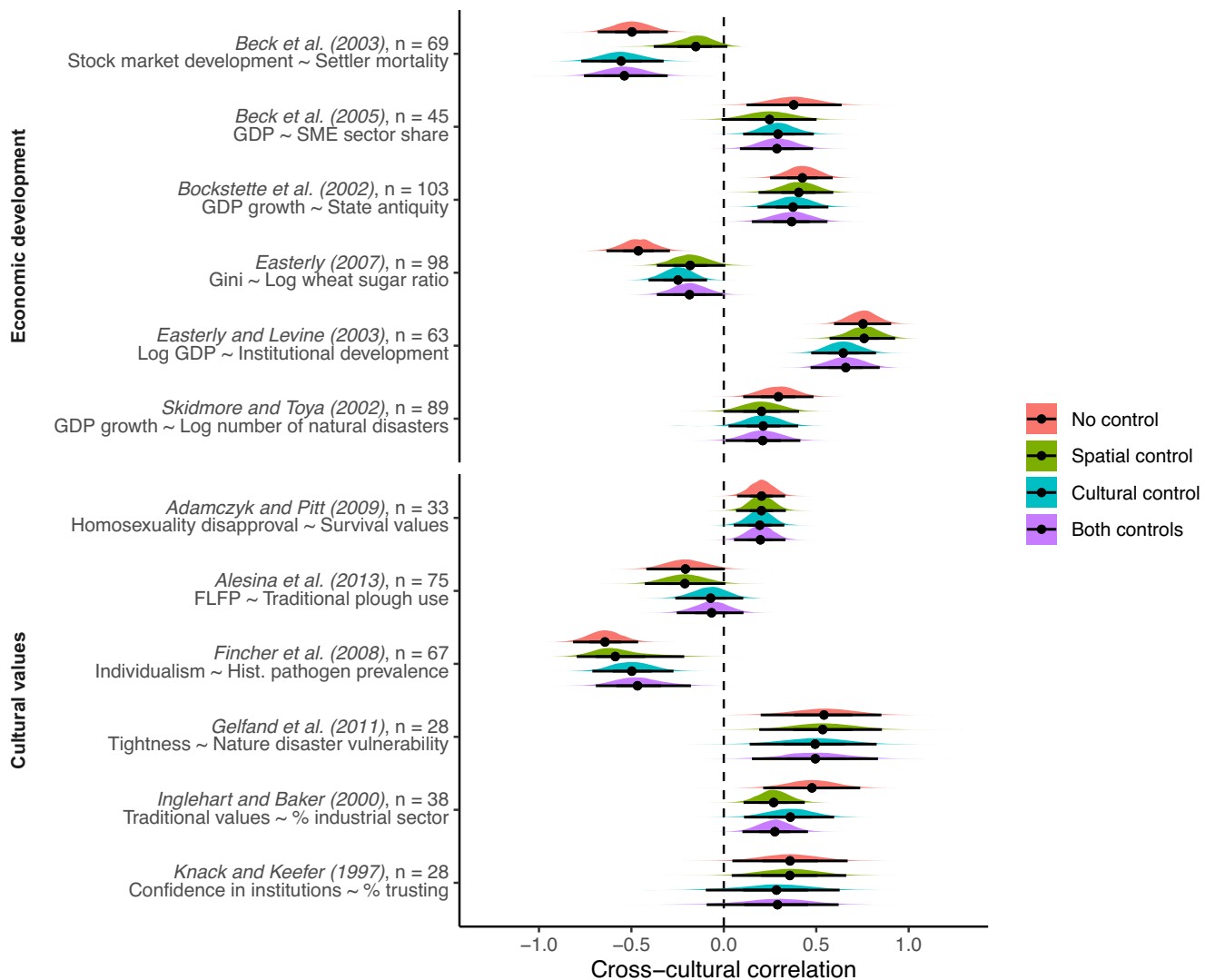

**Fig. 5 | Posterior correlations from our reanalysis of 12 previous cross-national analyses.** For each previous cross-national relationship, we plot the posterior slopes from a naive regression (red), a regression controlling for spatial non-independence (green), a regression controlling for cultural phylogenetic non-independence (blue), and a regression controlling for both spatial and cultural phylogenetic non-independence simultaneously (purple). All outcome and predictor variables are standardised. Most analyses are simple bivariate cross-national correlations, but Gelfand et al. (2011) is a partial correlation controlling for log gross national income, and Adamczyk and Pitt (2009) is a multilevel model including several covariates. Coloured ridges are full posterior distributions, and points and black lines represent posterior medians and equal-tailed 95% credible intervals. Numbers of observations from the models are as follows, from top to bottom: $n = 69$, $n = 45$, $n = 103$, $n = 98$, $n = 63$, $n = 89$, $n = 33$, $n = 75$, $n = 67$, $n = 28$, $n = 38$, and $n = 28$. GDP gross domestic product, FLFP female labour force participation.

(see "Methods")—and stopped when we had sampled a set of 12 analyses for which we were able to replicate the original result. The final set included six analyses from our economic development review[55–60] and six from our cultural values review[13,14,16,61–63] that had available data and were able to be replicated. We pre-registered this set before running any control models (https://osf.io/uywx8/). We controlled for non-independence by including (1) a Gaussian process allowing nation random intercepts to covary according to a geographic proximity matrix and/or (2) nation random intercepts that covaried according to a linguistic proximity matrix (see Supplementary Methods for full models).

Figure 5 visualises the results of our reanalysis (see Supplementary Table 4 for numerical results). Cross-national correlation effect sizes tended to reduce when controlling for statistical non-independence between nations, sometimes by as much as half of the original effect size. Overall, after controlling for non-independence, six out of twelve cross-national associations had 95% credible intervals that included zero. For the economic development analyses, four out

of six cross-national relationships had 95% credible intervals including zero when controlling for spatial non-independence. For the cultural values analyses, two out of six cross-national relationships had 95% credible intervals including zero when controlling for cultural phylogenetic non-independence. Supplementary Fig. 11 shows these cross-national correlations plotted against the raw data.

To understand why some cross-national correlations were attenuated by controls for non-independence while others were robust, we further explored our fitted models for evidence of spatial and cultural autocorrelation. For each outcome variable, our Gaussian process models provided varying estimates of how quickly spatial autocorrelation declined with distance (Supplementary Fig. 12). For example, in Skidmore and Toya[60] gross domestic product growth was only moderately spatially autocorrelated at 1000 km distance (posterior median spatial autocorrelation at 1000 km = 0.42, equal-tailed 95% CI [0.07 0.90]), whereas in Inglehart and Baker[16] traditional values were strongly spatially autocorrelated at the same distance (posterior median spatial autocorrelation at 1000 km = 0.96, 95% CI [0.81 0.99]).

We also found varying estimates of cultural phylogenetic signal (Supplementary Fig. 13), with some outcome variables expressing low signal (e.g., confidence in institutions[63]; posterior median = 0.07, 95% CI [0.00 0.53]) and others expressing high signal (e.g., female labour force participation[62]; posterior median = 0.89, 95% CI [0.63 0.98]). Exploratory regressions provided suggestive evidence that stronger estimates of spatial autocorrelation or cultural phylogenetic signal resulted in a more pronounced reduction in the effect size when controlling for non-independence between nations (Supplementary Fig. 14). However, these negative slopes for spatial autocorrelation (b = −0.19, 95% CI [−1.33 1.04]) and cultural phylogenetic signal (b = −0.28, 95% CI [−1.26 0.70]) were both very uncertain, due to the small number of analyses and the posterior uncertainty in effect sizes and estimates of non-independence.

## Discussion

In a literature review and simulation, we found that cross-national studies in economics and psychology rarely account for non-independence between nations, and when they do, the methods they use are insufficient to reduce false positives in non-independent data. In a reanalysis of 12 cross-national correlations, we further showed that neglecting to account for non-independence has resulted in potentially spurious relationships in the published literature, with half of the correlations failing to replicate when controlling for spatial or cultural non-independence with more rigorous methods. These findings suggest that cross-national analyses in economics and psychology should be interpreted with caution until non-independence is sufficiently accounted for.

Our initial analyses add to and clarify existing evidence regarding the degree of non-independence for national-level economic and cultural variables. One previous study suggested that geographic proximity is more important than deep cultural ancestry in explaining the distribution of human development across Eurasian nations, though the authors noted that their small sample of 44 nations and regional focus limited their statistical power[64]. By contrast, our global samples of over 160 nations revealed strong cultural phylogenetic signals, as well as geographic signals, for the Human Development Index, GDP per capita, and the Gini index of inequality. Another previous study found that similarities in the cultural values of nations are predicted by linguistic, but not geographic, distances between those nations[6]. We find this same result for survival vs. self-expression values, cultural tightness, and individualism, but for traditional vs. secular values, we find that both linguistic and geographic proximity are important independent predictors of global variation. These findings emphasise the need to account for both spatial and cultural phylogenetic non-independence in cross-national studies of economic development and cultural values.

Crucially, our literature review and simulation study revealed that the most commonly used controls for non-independence do not sufficiently deal with the issue. In our simulations, controlling for either latitude or longitude did not reduce false positive rates. This result calls into question the use of controls like distance to the equator to account for non-independence in cross-national regression models, though these controls may still be suitable to account for regional or latitudinal variation in climate, ecology, and natural threats (e.g., pathogens), which we did not simulate. High false positive rates persisted with Conley standard errors, despite recent claims that these standard error corrections are sufficient to deal with spatial non-independence[65]. The simulation also confirmed the assertion that fixed effects for spatial or cultural groupings (e.g., continent or language family fixed effects) are insufficient because non-independence still remains within groupings[43]. This logic further applies to analyses that control for non-independence by separately analysing different regions[66]. Controlling for the mean of the predictor variable within a 2000-km radius[48] eliminated false positive rates in spatially

autocorrelated data but had reduced statistical power to detect true associations. Across all model types in our simulation, the only methods that reduced false positive rates while retaining high statistical power were the random effects models with covariance matrices. The important advantage of these models is not that they are Bayesian per se (any of the approaches used here could be implemented in a Bayesian framework) but rather that they explicitly model covariance as a function of spatial or cultural distance.

There are other approaches to controlling for non-independence that we did not include in our simulation. For example, conditional autoregressive models[34] and generalised additive models[67] are approaches that can be applied in both frequentist and Bayesian frameworks. There are also alternative ways to operationalise cultural distances between nations beyond linguistic distances, including metrics like cultural fixation indices (e.g., cultural $F_{ST}$[6,68]), covariance based on genetic distances[69], and phylogenetic distances between religious traditions. We see merit in each of these approaches, and the use of one over another will depend on the specific question at hand. We decided to focus on linguistic distances in this study since language is a tangible socially learned trait that has previously been used to successfully track the effects of deep cultural ancestry on modern national outcomes[64,70,71]. Future work should explore whether other approaches are sufficient to reduce false positive rates in spatially and culturally non-independent data. In addition to spatial proximity and shared cultural ancestry, we did not simulate other sources of non-independence that potentially exist in real cross-national datasets, such as modern connections between nations due to flows of people and information (e.g., flight networks, social media networks) and shared histories of colonialism and capitalist modes of production that have shaped today's global landscape[72,73]. Additional controls will be required to ensure that these sources of non-independence do not confound cross-national inferences.

Ours is not the first review to show that studies are misapplying statistical methods in ways that inflate false positive rates. For example, other literature reviews have shown that studies in the social sciences tend to use small samples of participants[74], treat ordinal data as metric[75], incorrectly handle missing values[76], and ignore best practices in meta-analyses[77]. Why do cross-national studies also rarely account for non-independence? At the institutional level, one possibility is that such practices are incentivised because they generate statistically significant relationships, which increase the probability that a study is published[74]. Indeed, we found that controls for non-independence were less common among articles published in high-impact journals, suggesting that researchers are rewarded for such practices. At the individual level, another possibility is that researchers outside of anthropology and ecology are less aware of the problem or believe that the problem does not apply to analyses of nations. Even if researchers appreciate the problem, they might not know of suitable controls or perceive the methods to be too complex.

These institutional- and individual-level barriers can be combatted. First, cross-national replication studies like ours and others[42–46], combined with the methodological reviews included in Registered Reports[78], might change incentive structures and encourage researchers to analyse the world's nations with more rigorous methods. Second, more explicit descriptions of causal models could promote controls for non-independence by clearly outlining the nature of confounding and the sources of autocorrelation in cross-national data[79]. The causal model outlined in Supplementary Fig. 1 is a useful example, but individual studies must outline their own particular causal assumptions, which may include further sources of non-independence and confounding variables to control for (e.g., post-communist status, colony status). These causal models can then be used to design tailored statistical estimation strategies. Indeed, in our review, economists studying economic development dealt with national-level non-independence more than psychologists studying

cultural values, likely because economics studies tend to be lengthy statistical exercises that systematically incorporate or exclude numerous variables in an attempt to infer causation. Third, the recent widespread accessibility of open-source statistical software, such as the programming language Stan[80] and the R package brms[81], should promote the use of more rigorous methods to control for non-independence. Using brms, for example, Bayesian Gaussian process regression is straightforward to conduct, requiring only longitude and latitude values for nations. We have provided an online tutorial to help researchers apply these methods to their own cross-cultural datasets (https://scottclaessens.github.io/blog/2022/crossnational/).

Until such changes are implemented and sufficient controls for non-independence are the norm, existing cross-national correlations should be interpreted with caution. In our reanalyses, we found that half of the cross-national correlations had equal-tailed 95% credible intervals that included zero when controlling for spatial and/or cultural phylogenetic non-independence. While these results are in line with previous reanalyses[42], we note that we are unable to outright reject the claims from these studies since we only reanalysed the first bivariate regression specifications presented in the papers[65]. More detailed sets of reanalyses would be required to comprehensively challenge the claims from these specific papers. Nevertheless, these reanalyses do show, more broadly, that the problem of statistical non-independence applies to a wider range of national-level variables than those identified by previous work, such as parasite stress and democratic outcomes[43]. Moreover, given our finding that most studies in the current cross-national literature do not deal with non-independence at all, our reanalyses raise the worrying possibility that this literature is populated with spurious relationships. Future work should expand our set of reanalyses to determine the extent of this problem in the literature.

We do not wish to dissuade researchers from conducting cross-national studies. On the contrary, such work promises to deepen our understanding of our world, including the causes and consequences of economic development and cultural values. Moreover, cross-national studies allow social scientists to broaden their scope of study beyond Western populations[7], providing the representative samples necessary to test evolutionary and socio-ecological theories of human behaviour[8,82]. But in order to minimise spurious relationships in global datasets, we urge researchers to control for spatial and cultural phylogenetic non-independence when reporting cross-national correlations. Nations are not independent, and our statistical models must reflect this.

## Methods

### Geographic and cultural phylogenetic signal for measures of economic development and cultural values

To estimate the degree of spatial and cultural phylogenetic non-independence in economic development and cultural values, we calculated geographic and cultural phylogenetic signals for global measures of development and values. For economic development variables, we retrieved longitudinal data on the Human Development Index[47] (1990–2019; $n = 189$ nations), gross domestic product per capita (1960–2021; $n = 209$ nations), annual percentage growth in gross domestic product per capita (1961–2021; $n = 208$ nations), and the Gini coefficient of income inequality (1967–2021; $n = 167$ nations). Human development data were retrieved from the United Nations Development Programme (https://hdr.undp.org/en/content/download-data), and data for all other economic development variables were retrieved from the World Bank (https://data.worldbank.org/). For cultural values variables, we retrieved longitudinal data on traditional vs. secular values and survival vs. self-expression values from the World Values Survey[16] (1981–2019; $n = 116$ nations). We downloaded the full Integrated Values Survey, which included all waves from the World Values Survey and the European Values Survey, and computed the two dimensions of cultural values following procedures from previous research[16]. Additionally, we

retrieved cross-sectional data on cultural tightness ($n = 57$ nations) and individualism ($n = 97$ nations) from previous work[13,83].

To calculate geographic and cultural phylogenetic signals, we created two proximity matrices for 269 of the world's nations: a geographic proximity matrix and a linguistic proximity matrix. The geographic distance between two nations was calculated as the logged geodesic distance between nation capital cities (data from the maps R package[84]) using the geosphere R package[85]. The geographic proximity matrix was computed as one minus the log geographic distance matrix scaled between 0 and 1. Linguistic proximity between two nations was calculated as the cultural proximity between all languages spoken within those nations, weighted by speaker percentages. We acquired cultural proximity data by combining the language family trees provided by Glottolog v3.0[86] into one global language tree (undated and unresolved). We calculated cultural proximity $s$ between two languages $j$ and $k$ as the distance (in number of nodes traversed) of their most recent common ancestor $i$ to the root of the tree through the formula:

$$s_{jk} = \frac{n_r - n_i}{n_r} \tag{1}$$

where $n_r$ is the maximum path length (in number of nodes traversed) leading to the pan-human root $r$, and $n_i$ is the maximum path length leading to node $i$. We then combined these proximities with speaker data from Ethnologue 21[87] and compared every language spoken within those nations by at least 1 permille of the population, weighted by speaker percentages, through the formula:

$$w_{lm} = \Sigma\Sigma p_{lj} p_{mk} s_{jk} \tag{2}$$

where $p_{lj}$ is the percentage of the population in nation $l$ speaking language $j$, $p_{mk}$ is the percentage of the population in nation $m$ speaking language $k$, and $s_{jk}$ is the proximity measure between languages $j$ and $k$[88]. This calculation resulted in a linguistic proximity matrix with values between 0 and 1.

We included these matrices in Bayesian multilevel models, allowing nation random intercepts to covary according to both geographic and linguistic proximity simultaneously. These models were fitted with the R package brms[81] and converged normally ($\hat{R} < 1.1$). The assumptions of these models were met: residuals were approximately normally distributed, though this was not formally tested. Estimates of geographic and cultural phylogenetic signals were computed as the proportion of national-level variance in these models explained by geographic and linguistic proximity matrices.

### Literature review

We exported two searches from Web of Science (https://www.webofknowledge.com/) on 27 September 2021, restricting our searches to articles published between 1900 and 2018. The first search was for the terms "economic development" AND ("cross-national" OR "cross-cultural" OR "cross-country"), which returned 965 articles. The second search was for the terms "values" AND ("cross-national" OR "cross-cultural" OR "cross-country"), which returned 6806 articles. As this was not a formal systematic literature review, we did not follow PRISMA[89] guidelines for systematic literature reviews.

Once exported, we ordered the articles by descending number of citations per year since initial publication, using citation counts reported by Web of Science. We then coded each article in order for inclusion in our review. Articles were only included if: (1) they were judged to be relevant to economic development or cultural values; (2) they were an original empirical research article; and (3) they contained at least one analysis with national-level outcome or predictor variables. We stopped when we had included 50 articles for the economic development review and 50 articles for the cultural values review.

Within each included article, we exhaustively coded every individual cross-national analysis reported in the main text. We coded mainly correlation or regression analyses and explicitly excluded meta-analyses, factor analyses, measurement invariance analyses, multidimensional scaling analyses, hierarchical clustering analyses, multiverse analyses, and scale development/validation analyses. We also excluded analyses that compared only two, three, four, five, or six nations. For each included analysis, we recorded the year, impact factor of the journal (retrieved from https://jcr.clarivate.com/jcr/home), outcome variable, all predictor variables, test statistic, $p$-value, number of nations, number of data points, model type, if the data were available, and whether and how the analysis attempted to control for non-independence.

We coded common attempts to control for non-independence between nations. These included: (1) any higher-level control variables for spatial regional groupings (e.g., continent fixed effects); (2) any geographic distance control variables (e.g., distance between capital cities, distance from equator, latitude); (3) any control variables capturing shared cultural history (e.g., former colony, legal origin fixed effects, linguistic history, cultural influence); and (4) any other control variables, tests, or approaches that were deemed as attempts to control for non-independence (e.g., eigenvector filtering[90], controls for trade-weightings between nations, cross-sectional dependence tests[91], separate analyses for subsets of nations). These were coded by the first author.

Once we had compiled our review database, we calculated the proportion of articles attempting to control for non-independence at least once. We also calculated the proportion of articles employing the different types of control listed above at least once: regional fixed effects, distance, shared cultural history, or other. For these proportions, we calculated two-tailed 95% bootstrap confidence intervals with 1000 bootstrap iterations. Additionally, we predicted the probability of an article attempting to control for non-independence at least once using Bayesian logistic regression, including in separate models log journal impact factor and year of publication as linear and spline predictors, respectively.

For individual analyses, we dealt with the nested nature of the data (analyses nested within articles) by fitting Bayesian multilevel logistic regression models with review type (economic development vs. cultural values) as the sole fixed effect and random intercepts for articles. We fitted these models separately for overall attempts to control for non-independence and split by method type. We report the adjusted proportions with equal-tailed 95% credible intervals (see "Results" and Supplementary Fig. 2). Additionally, we predicted the probability of an analysis attempting to control for non-independence using Bayesian multilevel logistic regression with random intercepts for articles. In separate models, we included log journal impact factor and year of publication as linear and spline predictors, respectively. All Bayesian models were fitted with the brms R package[81]. Our priors were informed by prior predictive checks, and all models converged normally ($\hat{R} < 1.1$). The assumptions of these models (i.e., binary nested data) were met.

**Simulations**

We simulated data for 236 nations $i$ with varying degrees of spatial or cultural phylogenetic signal for outcome $y$ and predictor $x$ using the following generative model:

$$\begin{bmatrix} y_i \\ x_i \end{bmatrix} \sim \text{MVNormal}\left(\begin{bmatrix} \alpha_y \\ \alpha_x \end{bmatrix}, \mathbf{S}\right)$$
$$\alpha_y \sim \text{Normal}(0, \sqrt{\lambda} \cdot \Sigma)$$
$$\alpha_x \sim \text{Normal}(0, \sqrt{\rho} \cdot \Sigma) \qquad (3)$$
$$\mathbf{S} = \begin{pmatrix} \sqrt{1-\lambda} & 0 \\ 0 & \sqrt{1-\rho} \end{pmatrix} \begin{pmatrix} 1 & r \\ r & 1 \end{pmatrix} \begin{pmatrix} \sqrt{1-\lambda} & 0 \\ 0 & \sqrt{1-\rho} \end{pmatrix}$$

where $\Sigma$ is a correlation matrix proportional to either geographic or linguistic proximities between nations, $\lambda$ and $\rho$ are autocorrelation parameters that represent the expected spatial or cultural phylogenetic signal for outcome and predictor variables, respectively, and $r$ is the true cross-national correlation between the variables after accounting for autocorrelation. Importantly, when $r = 0$ in this simulation, we know that there is no direct causal relationship between $y$ and $x$. Instead, any relationship between the two variables is merely the result of autocorrelation.

We set the autocorrelation parameters $\lambda$ and $\rho$ to either 0.2 (weak), 0.5 (moderate), or 0.8 (strong). We also initially set the true cross-national correlation to 0 in order to determine false positive rates and then additionally set $r$ to 0.1 (small effect), 0.3 (medium effect), and 0.5 (large effect) in order to determine statistical power to detect true effects. For each parameter combination, we simulated 100 datasets, resulting in 3600 datasets. Each dataset had 236 rows representing different nations, with the following associated data for each nation: latitude, longitude, continent (Africa, Asia, Europe, North America, Oceania, or South America), language family of the nation's majority-spoken language (Afro-Asiatic, Atlantic-Congo, Austroasiatic, Austronesian, Eskimo-Aleut, Indo-European, Japonic, Kartvelian, Koreanic, Mande, Mongolic-Khitan, Nilotic, Nuclear Trans New Guinea, Sino-Tibetan, Tai-Kadai, Tupian, Turkic, or Uralic), the mean of the predictor variable within a 2000-km radius, and coordinates for genetic distances from a previous study[19] (only available for 177 nations).

With the resulting simulated datasets, we standardised outcome and predictor variables and fitted 11 different models: (1) naive regression without controls, (2) regression with latitude control, (3) regression with longitude control, (4) regression with continent fixed effects, (5) regression with language family fixed effects, (6) regression controlling for the mean of the predictor variable in a 2000-km radius, (7) regression employing Conley standard errors based on geographic distances, (8) regression employing Conley standard errors based on genetic distances, (9) Bayesian regression including a Gaussian process over latitudes and longitudes, (10) Bayesian regression including random intercepts covarying according to linguistic proximity, and (11) Bayesian regression including both a Gaussian process over latitudes and longitudes and random intercepts covarying according to linguistic proximity.

Models employing Conley standard errors either required latitude and longitude values or coordinates for genetic distances. To determine distance cutoffs, we employed an approach recommended in previous work[92]: we fitted models with a range of feasible distance cutoffs and retained the model with the largest standard error for the slope parameter. These models were fitted using the conleyreg R package[93]. Bayesian models were fitted using the brms R package[81]. Our choice of priors was based on prior predictive simulation. All models converged normally ($\hat{R} < 1.1$). Across all model types and parameter combinations, we calculated the false positive rate as the proportion of models that estimated slopes with a two-tailed 95% confidence/credible interval excluding zero when $r = 0$. We calculated statistical power as the proportion of models that estimated slopes with a 95% confidence/credible interval excluding zero when $r > 0$. We calculated two-tailed 95% bootstrap confidence intervals for these false positive rates and statistical power estimates with 1000 bootstrap iterations.

**Reanalyses**

We searched the individual analyses from our literature review for statistically significant cross-national correlations with available primary or secondary data. We restricted our search to one analysis per paper and searched until we had a set of 12 analyses, six from economic development papers and six from cultural values papers, for which we were able to replicate the original result (i.e., find a cross-national

correlation with the same sign and roughly the same effect size). We also ensured that at least one analysis was a multilevel model, with multiple observations per nation.

The 12 analyses that we settled on[13,14,16,55–63] were mostly bivariate cross-national correlations, except for two. One analysis[14] additionally controlled for log gross national income, and another analysis[60] is a multilevel model including random intercepts for nations and several individual-level and national-level covariates (see Model 5 in the original paper). Before running any additional models, we pre-registered these 12 analyses on the Open Science Framework on 25th January 2022 (https://osf.io/u8tbf). We endeavoured to keep the sample sizes of our reanalyses as close to the original analyses as possible, though there were some deviations (see Supplementary Table 5). Despite these slight deviations from the original analyses, all models reported in Fig. 5 are fitted to the same number of data points, meaning that any changes in effect sizes are solely due to controlling for non-independence.

For each individual analysis, we ran four models: (1) a naive regression replicating the original finding, (2) a regression including a Gaussian process allowing nation random intercepts to covary according to a geographic proximity matrix from latitude and longitude values, (3) a regression including nation random intercepts that covaried according to a linguistic proximity matrix, and (4) a regression including both a geographic Gaussian process and nation random intercepts with linguistic covariance. See Supplementary Methods for full models.

We fitted these models using the brms R package[81]. Our choice of priors was based on prior predictive simulation. All models converged normally ($\hat{R} < 1.1$), though for some models, we resorted to using approximate Gaussian processes[94] to reach convergence. The assumptions of these models were met: residuals were approximately normally distributed, though this was not formally tested.

### Reproducibility

All data and code are accessible on GitHub[95]. We used the targets R package[96] to create a reproducible data analysis pipeline and the papaja R package[97] to reproducibly generate the manuscript.

### Reporting summary

Further information on research design is available in the Nature Portfolio Reporting Summary linked to this article.

## Data availability

All data to reproduce the statistical analyses in this manuscript can be found on GitHub[95]. Data on human development were retrieved from the United Nations Development Programme (https://hdr.undp.org/en/content/download-data). Data on GDP per capita, annual GDP per capita growth, and the Gini coefficient were retrieved from the World Bank (https://data.worldbank.org/). Data on traditional vs. secular values and survival vs. self-expression values were retrieved from the World Values Survey (https://www.worldvaluessurvey.org/wvs.jsp). Data on cultural tightness were retrieved from the OSF repository for ref. 83 (https://osf.io/47pe8/). The review data generated in this study are provided in Supplementary Data 1. All other datasets (e.g., for replications) were retrieved from tables and Supplementary Tables directly from papers cited in the main text[13,14,16,55–63]—these datasets can be found in our GitHub repository[95].

## Code availability

All code to reproduce the statistical analyses in this manuscript can be found on GitHub[95].

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

## Acknowledgements
This work was supported by a Royal Society of New Zealand Marsden grant (20-UOA123) to Q.D.A.

## Author contributions
S.C. and Q.D.A. conceived of and designed the study. S.C. curated the data, produced all code for analysis and visualisation, and wrote the original draft of the manuscript. T.K. and Q.D.A. developed and compiled the geographic and linguistic distance matrices. Q.D.A. provided funding and input on manuscript preparation and revision. All authors reviewed and edited the final draft of the manuscript.

## Competing interests
The authors declare no competing interests.

## Ethics
We did not apply for ethical approval through an ethics board as we analysed only publicly available and simulated data in this study.
