## [Peer Review File · Nature Communications]

Cross-national analyses require additional controls to account for the non-independence of nationsREVIEWER COMMENTS

Reviewer #1 (Remarks to the Author):

I think this is a really important paper, and it could go a long way towards establishing geographic and linguistic interdependence controls as a norm in cross-cultural research. As a psychologist who studies culture, I think these controls are sorely needed in my field, and I found this paper to be a clear and helpful resource for my own research. Well done to the authors for such excellent work.

I have several low-level suggestions which may improve the paper. My general suggestion is that the paper might be strongest if it (a) moves its first result to the supplement since prior research has already established spatial and phylogenetic covariance in economic indicators and values, and (b) fleshes out its remaining sections by adding more descriptive text, statistics, and resources. Here are more specific suggestions for each section:

Abstract

The phrase "When studies do include controls for non-independence, our simulations suggest that commonly used methods continue to produce false positives" is unclear. I think what you mean to say is that common controls for non-independence are insufficient for reducing false positives in spatially autocorrelated data.

Introduction

You make a point in the second paragraph of your introduction that lots of articles (80k+) use terms like "cross-national" and "cross-cultural," but I am not compelled much by this point because it lumps together all the terms, and most of these articles could use the term "cross-cultural" and sample ethnic groups within a nation or societies from the Ethnographic Atlas or SCCS, which are not the focus of this paper. It would be more convincing to sample n papers that use the term "cross-cultural" and then to show that lots of these papers use nations as units of analysis.

First analysis

I thought that this was the weakest analysis of the paper. The three values seem quite arbitrary in a sea of economic and cultural variables that researchers use in their analyses. Moreover, the analyses don't tell us the scope of non-independence. In multi-level models, an ICC statistic will tell you that XX% of variance is explained by its group membership, which is a nice heuristic for researchers to use when determining whether they are dealing with non-independent datapoints. My suggestion would be to either expand the scope of this analysis and provide concrete metrics of non-independence, or to move the analysis to the supplement and cut to the more ambitious analyses that follow. You are not winning hearts and minds with this analysis anyway. Few researchers would argue that the HDI is independent across nations.

Second analysis

I like the paper-level analysis, but I don't like the analysis-level analysis because most papers do (and should) include many analyses which show how the main result(s) change with different constellations of control variables, including some zero-order models which don't include controls and simply illustrate effect sizes. Moreover, papers may include analyses that do not require spatial autocorrelation because they do not assume independence of residuals (e.g., Kendall correlations) or because they are simply descriptive notes to the reader without underlying claims about causality which require rigorous control structures (e.g., wealth tends to be more unequally distributed in poor nations). My point is that claims like "5% [of analyses] are estimated to control for non-independence" are striking, but they do not mean that "5% of analyses that need to control for non-independence do

control for non-independence" if that makes sense. I personally would understand if a paper wanted to avoid jumping right into more complex models, so they build up controls in a stepwise fashion to show how results are robust in each step. But your paper's coding criteria would suggest that this paper did not rigorously test for Galton's problem.

I would recommend always providing statistics when relevant (throughout the ms). For example, instead of saying "has remained low since 1993" when describing the time trend, can you actually provide the slope estimate of time against controlling for non-independence?

Third analysis

Could you briefly explain to readers what "linguistic proximity matrices" means before you present the results? I know you touch on it in the main text and supplement, but many cross-cultural researchers will be unfamiliar with phylogenetic methods from linguistics and biology so they will need as much hand-holding as possible, especially if you want this paper to make a big impact beyond the usual circles of scholars who care about this stuff

When setting the parameters for "varying degrees of spatial or cultural phylogenetic autocorrelation," perhaps you could use real-world estimates of autocorrelation from your initial analysis (where you estimate autocorrelation in real-world values and economic indicators). This means that your rate of autocorrelation is constrained by real-world data rather than straying into unreasonable rates of autocorrelation (every nation is identical). Along these lines, can you define what "strong spatial autocorrelation" means? Does this mean that more than 50% of variation between nations is explained by geographic or phylogenetic interdependence? More than that?

I would recommend putting the results from this section into a table so that readers can compare and contrast different control strategies.

It's cool to see that "Bayesian spatial Gaussian process regression with longitude and latitude values outperforms all other methods." However, I can imagine some students of cross-cultural psychology getting intimidated by this finding because they have not been trained in Bayesian methods and don't know where to start learning these methods. Can you provide annotated code for these readers to adapt in R for their own studies? Or alternatively, suggest the best combination of controls for frequentist statisticians? In general, I think it's great when a paper pointing out a problem also provides practical solutions: public code or tutorials would be the most practical solution. We developed annotated accessible tutorials in R when we published our paper encouraging psychologists to use text analysis and comparative language methods last year (<https://osf.io/hvcg3/>)

It looks like, for spatial autocorrelation, "strong spatial autocorrelation" still returns a false positive rate of 17%, which is remarkably high. This makes me wonder again what "strong spatial autocorrelation" really means, but also whether researchers confronted with very high rates of spatial autocorrelation should apply additional corrections (e.g., more conservative alpha values) than just trying to control for autocorrelation with continent fixed effects or Bayesian regressions with longitude and latitude corrections. Alternatively, can they get the 17% number lower if they include multiple controls at the same time (it looks like your analysis focuses on one control at a time)?

Fourth analysis

I would recommend providing a table in addition to Figure 5 within the main text so that readers can see the effect sizes and confidence intervals of the studies that you re-analyzed before and after controlling for spatial autocorrelation. I have a hard time understanding the difference between the range of the black lines in Figure 5 (which seldom cross 0) and the range of the colored lines (which frequently cross 0), and I couldn't find the exact statistics from this analysis in either the main text or supplemental materials.

I am not familiar with some of the analyses in Figure 5, but I am familiar with Gelfand et al 2011. That said, the Gelfand analysis contained 33 nations and the re-analysis only contains 33. Why is this, and was the sample size reduced (compared to the original paper) in all of your re-analyses?

I hope these suggestions are helpful, and I also hope that this paper gets out into the world soon. Cross-cultural researchers need it!

Joshua Conrad Jackson

Reviewer #2 (Remarks to the Author):

Here are my comments:

1) I appreciate the review on studies that raise the issue of spatial and cultural dependence and how attempts to correct for these issues are important. But, I would give significantly more weight on studies that have been peer reviewed in reputable journals than others that have been circulating for years as working papers.

One example of a carefully executed and peer-reviewed study is that by Bromham, L., Hua, X., Cardillo, M., Schneemann, H. & Greenhill, S. J. on "Parasites and politics: Why cross-cultural studies must control for relatedness, proximity and covariation." published in the Royal Society Open Science 5, 181100 (2018).

Another study, however, is mentioned as follows. "One study replicated 25 analyses of "persistence" in economics, in which modern national outcomes are regressed against historical characteristics of those nations, and found that over half of the relationships were attenuated when controlling for spatial non-independence". When I tried to see where this prominent study is I saw that it was well mentioned on Twitter and intensely covered by the now infamous <https://www.econjobrumors.com/> but remaining a working paper the last 5 years. So, a blog sensation should not be used as a piece of credible research.

2) I am wondering how different is the current approach and findings compared to the work by Bromham et al on exactly the same topic published in the Royal Society Open Science 5 in 2018.

3) The way that the authors look at whether a study accounts or not for spatial/cultural dependence is looking at whether

an article uses regional fixed effects (e.g. continent fixed effects) to account for non-independence or controls for spatial distance

(e.g. latitude) and shared cultural history (e.g. colony status)."

I have not looked into each of the various papers individually but papers in economics over the last 20 years that exploit cross-national variation without controls for latitude, colonizer dummies, continental fixed effects would not be likely to be published in any decent journal.

So, which journals are these papers published in? Considerations of review quality are important. which are these 100 articles?

4) The authors state: "In addition, we included Conley standard errors, a widely used standard error correction that purportedly accounts for spatial non-independence". I was not sure what "purportedly" refers to. Do the authors believe that this method does not account for this. Do the authors claim that Conley is an improper procedure?

Reviewer #3 (Remarks to the Author):

The paper "The non-independence of nations and why it matters" is on an important topic – cultural and spatial non-independence of countries and how to address this non-independence in cross-country analyses. In simulations and a literature review, they show that this issue is only inadequately addressed in the literature and propose a Bayesian random effects model to address it. All this is highly interesting.

I have several points that may help to improve the paper (though I see none of this as a "deal breaker"):

(1) The simulation clearly shows the value of the Bayesian random effects framework and it contrasts it to other existing ways to address non-independence. I think, though, it would be great to learn a bit more about how the different methods (and the Bayesian RE) address non-independence. For example, Conley errors will leave the coefficients unchanged but just adjust the standard errors. Region fixed effects only exploit variation within those regions (and discard between region variation), while controlling for absolute latitude, well, just holds latitude constant and both approaches do not adjust the standard errors for non-independence. (I think discussing the last two points as addressing non-independence is quite generous...). Where does your new approach fall in relation to those other approaches? In addition to the controls, could/should the standard errors be adjusted in the Bayesian RE approach? Why have random intercepts and not also random slopes? Apart from being able to decrease the prob. of false positives, how good is Bayesian RE to estimate the true effects? Since phylogenetic regressions are commonly used it'd be interesting to learn how BRE differs and why it is better than phylogenetic regressions. Note, that all these points are no "deal-breaker" for the paper, I just think that this could strengthen it.

(2) Overall, I think in the social sciences there is relatively too much emphasize on spatial non-independence instead of cultural non-independence. The UK and the USA are geographically far apart but are historically- or culturally not independent. I therefore value the effort to also take cultural non-independence into account – this cool feature could be emphasized more. In the paper, this is done by using information on linguistic distance. Yet, linguistic distance is a very coarse. I think it is worthwhile to also build on genetic distance (for measure and its relation to cultural distance see Spolaore and Wacziarg, 2009). That is, the paper would be strengthened to investigate culture distance based on genetic distance (e.g. in addition to the linguistic one – a measure that is prob. better than linguistic distance).

(3) It'd be curious to see how Conley standard errors with genetic distances fare in the simulations. For example, Schulz (2022) and Schulz et al. (2019) calculated Conley standard errors based on genetic distance in addition to geographic distance. Since Conley is the choice of economists it may be interesting to have this included in the simulations.

(4) I recommend to also engage with the paper by Collela et al. (2019); this is a paper that the economic audience seem to follow at the moment to address spatial non-independence. In their paper, they suggest incrementally increasing distance cut-offs (with uniform spatial decay kernel) and then choosing the cut-off that maximizes the Conley standard errors. This could be used in addition or instead of the existing Conley standard errors in the simulation.

(5) I recommend to also engage with the arguments in the paper by Collela et al. (2019) and the paper by Voth (2021). This might make the economic audience more interested in this paper. I like that the paper does not over sensationalizes its findings. Though, maybe even better to state that some papers bolster their findings with additional analysis (e.g. sub-country, sub-region analysis etc...).

(6) Human development index and GDP are highly correlated. Yet, I kept asking myself why the authors chose the former over the latter. GDP is a substantially more common measure and as far as the paper is also addressed to economists it may make sense to put GDP results in the appendix?

(7) I found it generous that the authors counted latitude, continent fixed effects or colony status towards accounting for non-independence

(8) I'd recommend changing "nations" to "countries". Nation seems to be a political concept while country refers more to a regional political unit.

(9) I'd be curious to see how the procedure by Schulz (2022) fairs in the simulations. He added the mean of explanatory variables of the surrounding countries (e.g. those within 2000km range) to the

regressions. The regression thus picks up only local variation. I understand that this is specific and may not make it into the paper.

(10) The pick of the 12 paper that are investigated seemed a bit random. Certainly, there are more famous papers...

Enrico Spolaore, Romain Wacziarg (2009). "The Diffusion of Development", *The Quarterly Journal of Economics*, Volume 124, Issue 2, May 2009, Pages 469–529, <https://doi.org/10.1162/qjec.2009.124.2.469>

Schulz, Jonathan (2022). "Kin Networks and Institutional Development", *Economic Journal*

Fabrizio Colella, , Rafael Lalive, Seyhun Orcan Sakalli, and Mathias Thoenig (2019). "Inference with Arbitrary Clustering" Working paper

Joachim Voth (2021). "Persistence: Myth and Mystery", In Bisin and Federico, *Handbook of Historical Economics*

Reviewer #4 (Remarks to the Author):

This is an important paper. It presents evidence that nation states are not independent datapoints for social science research; that many published studies do not appropriately control for this non-independence; and that not controlling for this issue increases the false positive rate. While the problem of non-independence of nation states has been known for some time, in at least some social science disciplines, this paper provides a very clear, rigorous appraisal of this problem, adding new evidence that not controlling for the problem likely biases the academic literature. I recommend publication, and have only minor comments for the authors to consider.

The methods appear to be sound. This is quite a complex paper in that the authors present a series of analyses: (1) testing whether nation states can be considered independent in social science research; (2) investigating the published literature to determine whether controls for non independence are commonly used; (3) performing a simulation analysis showing that not controlling for non independence increases the false positive rate, highlighting a Bayesian method which controls for non independence and reduces the false positive rate down to reasonable levels, and showing that commonly used frequentist controls for non independence don't typically reduce the false positive rate down to acceptable levels; (4) re-running analyses for 12 published studies to show that the findings cannot be replicating in half of the studies when using methods which appropriately control for non independence. However, the analyses are clearly justified and explained, the methods section is well described, and the narrative flow of the paper is good.

My only comment about the methods is a request for more clarity in the final section of the paper – the re-analysis of published studies. The authors give limited information on how they chose 12 studies from the original 100, saying only that they choose studies where the data were available and they could replicate the original finding. Does this mean that for the other 88 studies chosen for earlier analysis, either the data could not be found or the original finding could not be replicated? If so, that seems noteworthy and could be mentioned in the paper. If instead, the authors didn't systematically try to replicate all 100 studies, but stopped when they had found 12 studies with available data where they could replicate the original finding, that could also be made clear. Were any other exclusion criteria used? For example, studies that did not find positive associations might have been excluded? This might also be useful for the reader to know.

I have a couple of small suggestions for additional discussion points:

1. It might be helpful to include some discussion of whether the 50% non-replication rate the authors found for published studies might approximately hold across all published studies which have used cross-national analysis. My guess might be that an even higher proportion of published studies would

not replicate, given that the studies which the authors were able to replicate might be more rigorous than other studies – assuming that studies in which data could be found and the original result replicated were higher quality than others. But this is perhaps getting into speculation the authors would not find helpful. This is just a soft suggestion for some additional discussion the authors might want to include.

2. As another suggested discussion point – can the authors provide any hope for the frequentists among us, that there might be frequentist controls that could reduce the false positive rate down to reasonable levels? Their conclusion is that Bayesian methods work better than frequentist for this problem – is this because there is something about Bayesian methods which is superior for the particular problem described here? If so, perhaps the authors could explain why. (I'm not recommending the authors include an essay on the merits of Bayesian versus frequentist stats here, just some insight into why Bayesian methods might have performed better in this particular instance).

My final comment is not one which is relevant to whether this paper should be published or not, but I wonder do the authors intend to (or have they already) contacted the authors of analyses whose significant findings they were not able to replicate? Or the editors of journals which published those papers? The analyses presented here suggest that those authors or editors should consider correcting or retracting the papers with non-replicable findings.

Reviewer #5 (Remarks to the Author):

The main finding of this work is that existing (economic and psychological) cross-national studies is unable to control for spatial and linguistic ("cultural") non-independence between nations.

As a socio-cultural anthropologist, I am overall sympathetic toward any attempt to highlight and systematically investigate (via the use of qualitative or quantitative methods) the potential importance of often overlooked features and variables in economists' and psychologists' comparisons between different social and cultural units or scales, including nations. But for the same reason, I am also skeptical towards any attempt (whether deliberate or as a result of theoretical sleight-of-hand) to reduce the differences and similarities between two or more such units to a mere question of spatial and/or cultural proximity. In particular, I find the exclusion of political-economic structures, traditions and practices (e.g. communist/post-communist versus liberal/neoliberal) to constitute a potential major problem in terms of the both the construct (what is being studied) and measurement (how it is being studied) validity of the present study. Consider, for example, the case of the former so-called Second World (the Soviet Union and its economic and political dependences around the world, including not just Eastern Europe but also Cuba and Vietnam, etc.). As been demonstrated by a number of prominent sociologists (e.g. Dunn, Burawoy, Ledeneva) and anthropologists (e.g. Yurchak, Verdery, Humphrey), routines and values embodied and institutionalized in the course of 2-3 generations of state socialism lived on (and still today live on) as a distinct post-socialist culture, which clearly needs to be taken into account when designing, operationalizing and (meta) data processing a study like the present one.

As already noted, I find the questions posed by and the conclusions from this study to be of potentially significant interest to scholars and students from across the social sciences, even if the nature of the approach as well as the data and the methods are such that it will mostly appeal to readers trained in quantitative, empirical/experimental traditions. On a related point, while the study seems to be well-grounded in and positioned with respect to the state of the start within behavioral science and evolutionary anthropology, it is quite thin when it comes to engaging with relevant work from sociology and, especially, social and cultural anthropology. To be sure, the authors cannot be expected to be in possession of any expert knowledge or overview of these neighboring fields, but given that they explicitly mention "anthropology" as a source of identity and inspiration (page 19, line 318), one

would expect them to reference at least a few major works from socio-cultural anthropology, which also engage with “non-independence between nations”, albeit from very different theoretical and methodological perspectives and approaches (think, for example, of Ferguson’s work on Africa, and Luhrmann’s comparative work on Christian belief and prayer).

In terms of data quality, clarity of the presentation and reproducibility, I have little or nothing to say – but then again, this could be due to the fact that I am quite unfamiliar with this genre (statistically driven meta-study) and field (behavioral science/evolutionary anthropology) of research. When respect to the question of validity of the approach, however, I have several questions and reservations. For one thing, I am (as already pointed out) quite concerned about the lack of political-economic constructs and variables in the study. Perhaps this point can best be illustrated with reference to Supplementary Figure S1: while I certainly agree with the authors that “geographic (G) and linguistic (L) relationships” need to be taken into consideration in cross-national research, why does the same not go for “political” (P) and “economic” (E) – or better still, political economic (PE) ones? In fact, the authors do mention so-called “institutional causes” (page 41) as a potential variable, but they only do so in passing and features pertaining to political-economic structures and traditions (e.g., collective land rights, non-liberal forms of governance, etc.) do not seem to play any role in the authors’ operationalization neither G or L. Indeed, and this is my second reservation, the author mostly seem to reduce cultural variation to linguistic ditto, and on the occasions when they do engage with extra-linguistic aspects of cultural life, they turn these into just two measures, “traditional vs. secular values” and “survival vs. self-expression values” (p. 21, line 364-370). It is true that the latter operationalization is justified with reference “to previous research” (ibid), but tellingly the research in question only amounts to a single article published in *American Sociological Review*! To repeat a previous point of mine, no-one could reasonably expect the authors do be able – or indeed willing – to engage in detail with the extremely extensive socio-cultural (as well as cultural sociological) literature pertaining to the complex, dynamic and interconnected ways in which cultural patterns, processes and dynamics are embedded within and influenced by political-economic structures and dynamics. However, at a minimum they should at least mention the existence of this literature and some of its main conclusions and insights, even if the purpose of doing so is to refute its relevance for the present study, or to criticize the overarching assumptions of such “holistic” and “contextual” approaches.

Given my background in socio-cultural anthropology, I do not have the technical competences to adequately evaluate any specific questions pertaining to the paper’s use of statistics and treatment of uncertainties. However, based on my experience from collaborating with computational social scientists, I am wondering whether one potential source of model bias come from the use of statistical R packages such as *brms* and *conleyreg*, which (as the authors also note on Page 37, line695-697) come with certain inbuilt and thus potentially black boxed assumptions.

In summary, I this is an interesting meta-study, whose results (and general message regarding the weaknesses with cross-national studies) are likely to be of interest to a wide range of scholars and students from the social, economic and behavioral sciences. However, further work needs to be done on the study’s theoretical constructs, as well as the operationalization of these variables into specific measures. In particular, the authors need to consider the importance of non-spatial and non-linguistic relationships (e.g. shared political history and culture), something which might best be accomplished by delving deeper into the relevant sociological and socio-cultural anthropology literature.

Reviewer #6 (Remarks to the Author):

In their manuscript “The non-independence of nations and why it matters”, the authors make a single point: Most cross-national analyses do not sufficiently control for the statistical covariance among nations that arises because of spatial and/or cultural proximity. They make this point extremely well. I

find the paper very clear, well written and (especially for a methods paper) really entertaining. All re-analyses were preregistered and the authors provide the full data and analysis code on GitHub (when trying to clone the repo, I encountered some problems with the "Integrated_values_surveys_1981-2021.sav" file, maybe the authors want to check if there is indeed some problem).

As requested by the editor, I will focus my comments mainly on the Bayesian modeling methods used here. The analyses are rigorous, clearly described and overall appropriate for the research question. I also appreciate the full mathematical model details in the supplementary information which greatly helped me evaluate their models. However, I also have some concerns and comments.

First, the authors seem to suggest that their phylogenetic/Gaussian process regression approach generally controls for the non-independence of nations. As described in Fig. S1, the problem this approach is addressing is that unobserved variables U influence both X and Y and thus confound the causal effect. Their approach tries to model the covariation among populations that arises from such unobserved confounds. However, this method is no panacea for causal inference and also makes strong assumptions about the nature of confounding and our ability to measure cultural proximity. Specifically, the method only sufficiently controls for U if geographic G and linguistic relationships L (as defined by the user) are indeed the only causes of U . In general, such methods will only work if researchers can find good proxies for U and the success will depend on how closely such proxies correspond to U , with spatial and linguistic proximity only being two candidate variables. In a different context, variables other than G and L might be more appropriate to model U , the general point being that researchers should explicitly model the co-variance among nations to take account of their non-independence. Additionally, the models assume that linguistic similarity accurately reflects the covariance of those cultural traits that influence the unobserved nation-level variables U . Of course, strong assumptions are always necessary for causal inference in observational settings (Rohrer, 2018), so this is not a problem per se, the authors should just make those assumptions more transparent to not overexaggerate the power of this method and make readers aware of its inferential limits.

Next, I find the way the authors use Gaussian processes to model spatial proximity very sensible and, thus, do not understand why they seem to use a much more rigid approach for cultural similarity where they assume a fixed, linear relationship between phylogenetic distance and covariance. As rightly pointed out by the authors, their approach is justified if "cultural traits evolve via Brownian motion along a language phylogeny". However, most cultural traits do not evolve neutrally, and also evolve at different rates in different parts of the tree (assuming that linguistic phylogenies describe the evolution of relevant cultural traits at all...). The authors might consider using Gaussian processes for this analysis as well to estimate the covariance function and also allow for non-linear relationships between phylogenetic distance and covariance (see McElreath, 2020, chapter 14.5.2.).

Figure 6 shows the effect of the estimated spatial/cultural phylogenetic signal on the reductions in effect size in their re-analyses. Both variables are model estimates not data, but the analysis seems to ignore this uncertainty and only include the posterior medians instead of the full posteriors. This is a problem as there is usually quite some uncertainty in Gaussian-process control parameters which is also evident from Fig. S5. I would suggest the authors to repeat this analysis with the full posteriors instead of only posterior means to accurately reflect the full uncertainty.

More broadly, the problem of non-independence between nations is just one potential pitfall in cross-national research and the authors might want to discuss links to wider concerns about causal inference in cross-national/cross-cultural research. Valid comparisons and inference in a cross-national setting requires that researchers make assumptions about the precise mechanisms by which populations might differ (in this example: distance and linguistic similarity are the only mechanisms that generate co-variance between X and Y) and how the data are generated, and design tailored estimation strategies based on those causal assumptions (see Deffner et al., 2022). As one example, Deffner et al. (2022) discuss the "proxy control" approach using phylogenetic distance that is used in this ms

(Fig. 7b), but also discuss fundamental issues of measurement, description and generalization.

Lastly, in several places (e.g., lines 209, 218, 222), the authors present their “Bayesian” approach as an alternative to “frequentist methods” which do not sufficiently control for non-independence. However, the question here is not really about Bayesian or frequentist estimation, but about the random-effects covariance structure among nations. Therefore, also Bayesian models which do not model how this covariance arises from relevant causes will be inappropriate. Related to that, in the abstract, the authors write that they show that half of results “are no longer significant after controlling for non-independence”, which is misleading as they are not using classic significance testing and only some model estimates substantially change compared to the original values even if some arbitrary HPDI now overlaps 0.

Cited literature

Deffner, D., Rohrer, J. M., & McElreath, R. (2022). A causal framework for cross-cultural generalizability. PsyArxiv. <https://doi.org/10.31234/osf.io/fqukp>.

McElreath, R. (2020). *Statistical rethinking: A Bayesian course with examples in R and Stan*. Chapman and Hall/CRC.

Rohrer, J. M. (2018). Thinking clearly about correlations and causation: Graphical causal models for observational data. *Advances in methods and practices in psychological science*, 1(1), 27-42.

Response to Reviewers

Reviewer 1

I think this is a really important paper, and it could go a long way towards establishing geographic and linguistic interdependence controls as a norm in cross-cultural research. As a psychologist who studies culture, I think these controls are sorely needed in my field, and I found this paper to be a clear and helpful resource for my own research. Well done to the authors for such excellent work.

We thank this reviewer for their positive comments on our manuscript.

I have several low-level suggestions which may improve the paper. My general suggestion is that the paper might be strongest if it (a) moves its first result to the supplement since prior research has already established spatial and phylogenetic covariance in economic indicators and values, and (b) fleshes out its remaining sections by adding more descriptive text, statistics, and resources.

Thank you for your suggestions. We have responded to these broad points by (a) adding further national-level variables to the first analysis, rather than moving this result to the supplement, and (b) including further descriptions, inferential statistics, and tables of numerical results where appropriate. We also now link to a blog post tutorial for the Bayesian methods. We explain these changes more clearly and address specific concerns in the sections below.

Here are more specific suggestions for each section:

Abstract

The phrase “When studies do include controls for non-independence, our simulations suggest that commonly used methods continue to produce false positives” is unclear. I think what you mean to say is that common controls for non-independence are insufficient for reducing false positives in spatially autocorrelated data.

This suggested wording is better. We were constrained in our initial wording by the tight abstract word limit (150 words). We have updated this sentence.

Introduction

You make a point in the second paragraph of your introduction that lots of articles (80k+) use terms like “cross-national” and “cross-cultural,” but I am not compelled much by this point because it lumps together all the terms, and most of these articles could use the term “cross-cultural” and sample ethnic groups within a nation or societies from the Ethnographic Atlas or SCCS, which are not the focus of this paper. It would be more convincing to sample n papers that use the term “cross-cultural” and then to show that lots of these papers use nations as units of analysis.

We agree that we want to focus on national-level analyses here. Rather than following the reviewer’s suggested approach, we have instead opted to reduce our search only to articles that use the term “cross-national” in titles or abstracts. This produces over 13,000 unique hits.

First analysis

I thought that this was the weakest analysis of the paper. The three values seem quite arbitrary in a sea of economic and cultural variables that researchers use in their analyses. Moreover, the analyses don’t tell us the scope of non-independence. In multi-level models, an ICC statistic will tell you that XX% of variance is explained by its group membership, which is a nice heuristic for researchers to use

when determining whether they are dealing with non-independent datapoints. My suggestion would be to either expand the scope of this analysis and provide concrete metrics of non-independence, or to move the analysis to the supplement and cut to the more ambitious analyses that follow. You are not winning hearts and minds with this analysis anyway. Few researchers would argue that the HDI is independent across nations.

We agree with the reviewer that the particular outcome variables we initially chose to focus on (HDI, traditional values, and survival values) are relatively arbitrary when set against the full range of economic and cultural variables that researchers include in their analyses. We have thus opted to expand this first analysis to incorporate further national-level outcome variables, focusing on variables that will later appear in our review and reanalyses. These variables are not intended to be a comprehensive list of all national-level variables included in cross-national research, but rather an illustrative set of example variables that are widely used in the literature.

We have updated the justification for this first analysis in the text. Few researchers argue that nations are entirely independent, but some may believe that the effects are weak enough to ignore. Our goal is to rebut this latter claim by quantifying the strength of non-independence for a range of widely-used variables. We see this as a necessary first step in our argument, and one which we would prefer not to move to the supplementary materials.

We disagree with the claim that the intra-class correlations in these analyses do not reflect concrete metrics of the strength of non-independence. In standard multilevel models, the intra-class correlation indeed reflects the proportion of variance that is explained by group membership (i.e., the proportion of variance that is between, rather than within, nations). But when random intercepts are allowed to covary according to pre-specified linguistic or geographic covariance matrices, as they are in these models, the intra-class correlations for those random intercepts represent the proportion of variance that is specifically explained by cultural or spatial non-independence, rather than group membership alone. Villemereuil and Nakagawa (2014) show how this statistic is akin to Pagel's λ in phylogenetic contexts (for an example in R, see here). As such, we believe that the size of the intra-class correlations from these analyses do reflect the strength of non-independence in the outcome variables.

Second analysis

I like the paper-level analysis, but I don't like the analysis-level analysis because most papers do (and should) include many analyses which show how the main result(s) change with different constellations of control variables, including some zero-order models which don't include controls and simply illustrate effect sizes. Moreover, papers may include analyses that do not require spatial autocorrelation because they do not assume independence of residuals (e.g., Kendall correlations) or because they are simply descriptive notes to the reader without underlying claims about causality which require rigorous control structures (e.g., wealth tends to be more unequally distributed in poor nations). My point is that claims like "5% [of analyses] are estimated to control for non-independence" are striking, but they do not mean that "5% of analyses that need to control for non-independence do control for non-independence" if that makes sense. I personally would understand if a paper wanted to avoid jumping right into more complex models, so they build up controls in a stepwise fashion to show how results are robust in each step. But your paper's coding criteria would suggest that this paper did not rigorously test for Galton's problem.

We are sympathetic to this point, especially since papers in economics often unfold in precisely this manner, beginning with simple analyses and slowly adding controls. In response to this comment, we have opted to move the analysis-level proportions to the supplementary materials, and we now discuss them only briefly in the main text. We think this result is still worth a mention in that the low proportion of analyses controlling for non-independence is striking nonetheless — if you read or hear of a cross-national relationship in the literature, it most likely will not have controlled for non-independence.

I would recommend always providing statistics when relevant (throughout the ms). For example, instead of saying “has remained low since 1993” when describing the time trend, can you actually provide the slope estimate of time against controlling for non-independence?

We now report the slopes of the time splines for both economic development and cultural values articles.

Third analysis

Could you briefly explain to readers what “linguistic proximity matrices” means before you present the results? I know you touch on it in the main text and supplement, but many cross-cultural researchers will be unfamiliar with phylogenetic methods from linguistics and biology so they will need as much hand-holding as possible, especially if you want this paper to make a big impact beyond the usual circles of scholars who care about this stuff

Yes, our initial wording assumed too much prior knowledge from the reader. We have expanded this section with a brief explanation of (1) how language phylogenies are constructed, (2) how the linguistic proximity matrix is constructed from language phylogenies, and (3) how the matrix can be understood as a proxy for cultural relatedness.

When setting the parameters for “varying degrees of spatial or cultural phylogenetic autocorrelation,” perhaps you could use real-world estimates of autocorrelation from your initial analysis (where you estimate autocorrelation in real-world values and economic indicators). This means that your rate of autocorrelation is constrained by real-world data rather than straying into unreasonable rates of autocorrelation (every nation is identical). Along these lines, can you define what “strong spatial autocorrelation” means? Does this mean that more than 50% of variation between nations is explained by geographic or phylogenetic interdependence? More than that?

We had attempted to do this already by comparing “weak”, “moderate”, and “strong” levels of autocorrelation in our simulations to the results from our first analysis. Regarding the proportion of variance explained, we have added a brief explanation in the main text clarifying that, for example, an autocorrelation parameter of 0.2 means that 20% of the national-level variance is explained by non-independence.

I would recommend putting the results from this section into a table so that readers can compare and contrast different control strategies.

We have added tables of simulation results in the supplementary materials. In the main text, we would prefer to present the simulation results as figures, since figures more easily convey the qualitative trends in the results than large tables of numbers.

It's cool to see that “Bayesian spatial Gaussian process regression with longitude and latitude values outperforms all other methods.” However, I can imagine some students of cross-cultural psychology getting intimidated by this finding because they have not been trained in Bayesian methods and

don't know where to start learning these methods. Can you provide annotated code for these readers to adapt in R for their own studies? Or alternatively, suggest the best combination of controls for frequentist statisticians? In general, I think it's great when a paper pointing out a problem also provides practical solutions: public code or tutorials would be the most practical solution. We developed annotated accessible tutorials in R when we published our paper encouraging psychologists to use text analysis and comparative language methods last year (<https://osf.io/hvcg3/>).

Shortly after submitting this manuscript, the first author produced an R tutorial on his blog so that these methods could be adopted by others (<https://scottclaessens.github.io/blog/2022/crossnational/>). In the revised manuscript, we now explicitly link to this blog post. The annotated code for the blog post, along with all the code for the analyses, simulations, and manuscript, is contained in a public GitHub repository.

Unfortunately, we are not aware of frequentist versions of the Bayesian Gaussian Process or correlated random effects regressions that we use in this manuscript. Other methods of dealing with non-independence, such as generalised additive models and phylogenetic least squares regression, can be conducted under frequentist frameworks, but we don't explore these models in the current manuscript.

It looks like, for spatial autocorrelation, "strong spatial autocorrelation" still returns a false positive rate of 17%, which is remarkably high. This makes me wonder again what "strong spatial autocorrelation" really means, but also whether researchers confronted with very high rates of spatial autocorrelation should apply additional corrections (e.g., more conservative alpha values) than just trying to control for autocorrelation with continent fixed effects or Bayesian regressions with longitude and latitude corrections. Alternatively, can they get the 17% number lower if they include multiple controls at the same time (it looks like your analysis focuses on one control at a time)?

We agree that, when confronted with high rates of autocorrelation, researchers could also utilise multiple controls at the same time (e.g., including local averages of predictor variables alongside a geographic covariance matrix) or additional corrections that we did not focus on (e.g., more conservative alpha values, generalised additive models, conditional autoregressive models, etc.). These combinations and additional corrections were beyond the scope of our particular simulation, but future simulation work should assess their efficacy. Regardless, our simulation results have shown that it is *not* defensible to ignore non-independence or only use individual controls that barely reduce false positive rates, such as language family fixed effects or longitude.

Fourth analysis

I would recommend providing a table in addition to Figure 5 within the main text so that readers can see the effect sizes and confidence intervals of the studies that you re-analyzed before and after controlling for spatial autocorrelation. I have a hard time understanding the difference between the range of the black lines in Figure 5 (which seldom cross 0) and the range of the colored lines (which frequently cross 0), and I couldn't find the exact statistics from this analysis in either the main text or supplemental materials.

The omission of numerical results was an oversight on our part. We have included a supplementary table with the slopes and 95% credible intervals from all of these reanalysis models. Regarding the figure, we have now edited the figure legend to explain that the

coloured densities (which produce coloured lines at the tails) are full posterior distributions, while the black lines represent 95% credible intervals.

I am not familiar with some of the analyses in Figure 5, but I am familiar with Gelfand et al 2011. That said, the Gelfand analysis contained 33 nations and the re-analysis only contains 33. Why is this, and was the sample size reduced (compared to the original paper) in all of your re-analyses?

Our reanalysis of the cross-national correlation in Gelfand et al. (2011) contains a slightly different number of data points ($N = 28$) due to missing data, repetition, and removal of a high leverage point. The original paper reports an analysis with $N = 30$, rather than the full dataset of 33 nations, because of missing data on predictors. In addition to this, we also combined both East and West Germany into a single nation to match our linguistic proximity matrix, and we removed Venezuela as a high leverage point, since it has a substantially higher natural disaster rate than the other nations. Despite these changes, our reanalysis is in line with the original paper in that we find a moderate-to-strong positive relationship between tightness and natural disaster vulnerability, and we additionally find that this relationship is robust to controls for spatial and cultural non-independence.

To ensure that deviations like this are not hidden from the reader, we have included a supplementary table that outlines the reasons for any variations between the original analyses and our reanalyses. However, despite any of these slight variations, the slopes that we compare in Figure 5 all come from analyses of the exact same datasets with the exact same number of data points, meaning that they are compared like for like.

I hope these suggestions are helpful, and I also hope that this paper gets out into the world soon. Cross-cultural researchers need it!

Joshua Conrad Jackson

Reviewer 2

Here are my comments:

1) I appreciate the review on studies that raise the issue of spatial and cultural dependence and how attempts to correct for these issues are important. But, I would give significantly more weight on studies that have been peer reviewed in reputable journals than others that have been circulating for years as working papers.

One example of a carefully executed and peer-reviewed study is that by Bromham, L., Hua, X., Cardillo, M., Schneemann, H. & Greenhill, S. J. on "Parasites and politics: Why cross-cultural studies must control for relatedness, proximity and covariation." published in the Royal Society Open Science 5, 181100 (2018).

Another study, however, is mentioned as follows. "One study replicated 25 analyses of "persistence" in economics, in which modern national outcomes are regressed against historical characteristics of those nations, and found that over half of the relationships were attenuated when controlling for spatial non-independence". When I tried to see where this prominent study is I saw that it was well mentioned on Twitter and intensely covered by the now infamous <https://www.econjobrumors.com/> but remaining a working paper the last 5 years. So, a blog sensation should not be used as a piece of credible research.

We disagree with the argument that working papers should not be cited as credible research. According to Google Scholar, the working papers from Morgan (2019) and Morgan (2020) have been cited 205 and 40 times, respectively. This work has even been cited by prominent economists in high-profile journals, such as Nathan Nunn's recent paper in *Science* (Nunn, 2020).

Of course, we understand that findings that have not yet gone through peer review should be treated with caution. To ensure that the reader is aware of the preliminary nature of these findings, we have clarified in the text that the paper is a working paper.

2) I am wondering how different is the current approach and findings compared to the work by Bromham et al on exactly the same topic published in the Royal Society Open Science 5 in 2018.

The Bromham et al. (2018) paper was one inspiration for our work, but that paper narrowly focuses on a specific case study — cross-national correlations between cultural traits and parasite load. Our work differs from Bromham et al. (2018) in a number of ways. First, we include a range of variables that are much more varied and apply broadly to research across the social sciences, including GDP per capita, GDP per capita growth, income inequality, individualism, cultural tightness, and traditional values. Second, we conduct a systematic review of the published literature to determine baseline rates of controls for non-independence in economics and psychology. Third, we conduct a simulation study to test whether commonly-used controls adequately deal with the issue of non-independence. In the Discussion section, we now explain how our results build on the work by Bromham et al. (2018) and others.

3) The way that the authors look at whether a study accounts or not for spatial/cultural dependence is looking at whether an article uses regional fixed effects (e.g. continent fixed effects) to account for non-independence or controls for spatial distance (e.g. latitude) and shared cultural history (e.g. colony status)." I have not looked into each of the various papers individually but papers in economics over the last 20 years that exploit cross-national variation without controls for latitude, colonizer dummies, continental fixed effects would not be likely to be published in any decent journal. So, which journals are these papers published in? Considerations of review quality are important. which are these 100 articles?

The articles from our review are the 100 articles with the highest annual rate of citations under our search terms from Web of Science. The review contains articles from prestigious journals such as *Science* (IF = 63.71), *Quarterly Journal of Economics* (IF = 19.01), *American Economic Review* (IF = 10.54), *American Sociological Review* (IF = 6.37), *Psychological Science* (IF = 4.90), and the *Journal of Economic Growth* (IF = 4.80), the majority of which were published in the last 20 years. In an additional analysis, we find that journal impact factors are actually *negatively* related to the probability of controlling for non-independence, at both the paper-level and the analysis-level (Figure 2b). We now report and discuss this result in the main text.

This comment made us realise that a full list of the papers considered in the review was not reported in the previous version of the manuscript, only in the raw data for the review itself. To remedy this, we now report a supplementary table listing the 100 papers included in the review.

4) The authors state: "In addition, we included Conley standard errors, a widely used standard error correction that purportedly accounts for spatial non-independence". I was not sure what

"purportedly" refers to. Do the authors believe that this method does not account for this. Do the authors claim that Conley is an improper procedure?

The reason that we used the word "purportedly" is that, in the simulation results that followed this sentence, we found that Conley standard errors did little to change the false positive rate under various levels of autocorrelation. Using three different reasonable distance cutoffs, we found that the Conley procedure continued to produce spurious relationships between outcome and predictor variables. Based on this finding, we concluded that the Conley procedure does not adequately deal with spatial (or cultural) non-independence, a conclusion that we continue to draw in the revised version of the manuscript. However, due to a rewrite of this section, the word "purportedly" has now been removed in the revised version of the manuscript.

Reviewer 3

The paper "The non-independence of nations and why it matters" is on an important topic – cultural and spatial non-independence of countries and how to address this non-independence in cross-country analyses. In simulations and a literature review, they show that this issue is only inadequately addressed in the literature and propose a Bayesian random effects model to address it. All this is highly interesting.

We thank this reviewer for their positive comments on our manuscript.

I have several points that may help to improve the paper (though I see none of this as a "deal breaker"):

(1) The simulation clearly shows the value of the Bayesian random effects framework and it contrasts it to other existing ways to address non-independence. I think, though, it would be great to learn a bit more about how the different methods (and the Bayesian RE) address non-independence. For example, Conley errors will leave the coefficients unchanged but just adjust the standard errors. Region fixed effects only exploit variation within those regions (and discard between region variation), while controlling for absolute latitude, well, just holds latitude constant and both approaches do not adjust the standard errors for non-independence. (I think discussing the last two points as addressing non-independence is quite generous...). Where does your new approach fall in relation to those other approaches? In addition to the controls, could/should the standard errors be adjusted in the Bayesian RE approach? Why have random intercepts and not also random slopes? Apart from being able to decrease the prob. of false positives, how good is Bayesian RE to estimate the true effects? Since phylogenetic regressions are commonly used it'd be interesting to learn how BRE differs and why it is better than phylogenetic regressions. Note, that all these points are no "deal-breaker" for the paper, I just think that this could strengthen it.

There are several points here. First, we agree that it would be good to provide more detailed explanations of how each method deals with non-independence (e.g., via standard error adjustment, discarding variation between regions, etc.). Our proposed Bayesian random effects models account for non-independence by modelling the covariance between nations that is induced by their geographic or linguistic connections. We have clarified this approach in the main text when introducing the different methods.

Second, we do not believe that standard errors should be adjusted in the Bayesian random effects models, because the inclusion of the covariance matrix is in effect already doing this

(i.e., the posterior slopes for any predictors, including their standard errors, are adjusted for non-independence within the model itself).

Third, the reason to only include random intercepts and not random slopes in the Bayesian random effects models is that the cross-national analyses we consider in our simulations have one data point per nation. As such, each nation cannot have its own “slope” for the predictor variable – there can only be a population-level slope. For analyses with multiple data points per nation, random slopes become possible, though they can be challenging to estimate (see here). For simplicity, we do not consider analyses of this kind in the manuscript.

Fourth, the reviewer asks how good the Bayesian random effects models are at detecting true effects. This is an important point. One could argue that these models are achieving low false positive rates because they are overly conservative, and in practice they might not have the power to detect true effects. To answer this question, we have expanded our simulation to include true effect sizes beyond zero. For effect sizes of varying strength (large = 0.5, medium = 0.3, small = 0.1), we test the power of each method. We find that, for large and medium effects, the Bayesian random effects models have consistently higher power than other methods at all levels of spatial and cultural autocorrelation. For small effects, all methods suffer from low power, likely due to the low sample size ($n = 236$). These power analysis results are reported in the supplementary materials.

(2) Overall, I think in the social sciences there is relatively too much emphasize on spatial non-independence instead of cultural non-independence. The UK and the USA are geographically far apart but are historically- or culturally not independent. I therefore value the effort to also take cultural non-independence into account – this cool feature could be emphasized more. In the paper, this is done by using information on linguistic distance. Yet, linguistic distance is a very coarse. I think it is worthwhile to also build on genetic distance (for measure and its relation to cultural distance see Spolaore and Wacziarg, 2009). That is, the paper would be strengthened to investigate culture distance based on genetic distance (e.g. in addition to the linguistic one – a measure that is prob. better than linguistic distance).

We agree that it is important to bring cultural non-independence to the attention of a wider social science audience. To emphasise this more, we have expanded our first description of cultural non-independence in the introduction section, describing how it differs from geographic distance but is equally as important in the study of nations.

Regarding the usage of genetic vs. linguistic distances, we acknowledge that some researchers have used measures of genetic distance to make claims about the role of cultural relatedness in shaping national-level outcomes (Guiso, Sapienza, & Zingales, 2009; Spolaore & Wacziarg, 2009, 2013, 2016). However, we prefer to focus on linguistic distance in this paper for a number of reasons. First, genetic distance is highly correlated with geography (Handley et al. 2007; Novembre et al. 2008) making it difficult to differentiate between the effects of cultural history and geographic patterns. Second, genetic distance is consistent with multiple underlying processes, such as deep cultural inheritance and more recent gene flow between populations, making it difficult to interpret associations with modern outcomes. By contrast, language ancestry and other cultural lineages track the genealogies of human populations based on known cultural markers. These measures are more likely to be distinct from geography and are more clearly tied to specific underlying processes (i.e., deep cultural inheritance).

This is not to say that linguistic distance is the “correct” proxy for cultural relatedness and genetic distance is not. Both are different approaches to the study of culture. There are of course other viable proxies, such as religious ancestry, which we have used in our previous work. But, for the reasons above, we have opted to use linguistic distance as our primary measure of cultural relatedness in the paper.

(3) It’d be curious to see how Conley standard errors with genetic distances fare in the simulations. For example, Schulz (2022) and Schulz et al. (2019) calculated Conley standard errors based on genetic distance in addition to geographic distance. Since Conley is the choice of economists it may be interesting to have this included in the simulations.

Researchers use genetic distance as a control for cultural non-independence in their regressions (Schulz, 2022; Schulz et al. 2019). To determine how well this method fares as a control for non-independence, we included Conley standard errors with genetic distances into our simulations. We find that false positive rates persist under this method, raising concerns about the usage of this method in the literature.

(4) I recommend to also engage with the paper by Collela et al. (2019); this is a paper that the economic audience seem to follow at the moment to address spatial non-independence. In their paper, they suggest incrementally increasing distance cut-offs (with uniform spatial decay kernel) and then choosing the cut-off that maximizes the Conley standard errors. This could be used in addition or instead of the existing Conley standard errors in the simulation.

Thank you for this valuable suggestion. The Collela et al. (2019) reference is particularly useful since we were not sure which distance cutoff to settle on when implementing Conley standard errors. We have implemented the “optimal correction threshold” approach in the simulation with a uniform spatial decay kernel, and again find that Conley standard errors continue to produce false positives.

(5) I recommend to also engage with the arguments in the paper by Collela et al. (2019) and the paper by Voth (2021). This might make the economic audience more interested in this paper. I like that the paper does not over sensationalizes its findings. Though, maybe even better to state that some papers bolster their findings with additional analysis (e.g. sub-country, sub-region analysis etc...).

We were not familiar with the Voth (2021) book chapter – thank you for the recommendation. It is informative to see how the claims from Kelly (2019) have been interpreted in the economics literature, and how we can potentially add to an ongoing debate. Upon reading this chapter, we have expanded our paragraph regarding “persistence” studies in the Discussion section, ultimately siding with Kelly (2019) over Voth (2021).

In response to the reviewer’s comment about papers bolstering their initial bivariate analyses with additional specifications, we now make clear up front that we are mostly reanalysing the *initial* specifications from the 12 papers.

(6) Human development index and GDP are highly correlated. Yet, I kept asking myself why the authors chose the former over the latter. GDP is a substantially more common measure and as far as the paper is also addressed to economists it may make sense to put GDP results in the appendix?

We have included an analysis of GDP as well as HDI. The results are similar for both variables.

(7) I found it generous that the authors counted latitude, continent fixed effects or colony status towards accounting for non-independence

To be conservative in our literature review, we wanted to include all reasonable “attempts” to control for non-independence (e.g., latitude, longitude, continent fixed effects, colony status) even though we agree that these attempts do not fully deal with the issue. The fact that over half of the papers in economics and psychology still do not include even these straightforward controls is all the more striking.

(8) I’d recommend changing “nations” to “countries”. Nation seems to be a political concept while country refers more to a regional political unit.

We would prefer to retain the word “nation”. Unlike labels like “country” or “state”, the label “nation” does not presuppose any particular form of political governance, and emphasises the deep cultural and historical ties between human populations that create global non-independencies.

(9) I’d be curious to see how the procedure by Schulz (2022) fairs in the simulations. He added the mean of explanatory variables of the surrounding countries (e.g. those within 2000km rage) to the regressions. The regression thus picks up only local variation. I understand that this is specific and may not make it into the paper.

We have included this method in our simulation. We find that this method is very effective at eliminating false positive rates at all levels of spatial autocorrelation, although it suffers from lower power than the Bayesian random effects models when detecting a medium effect size with strong spatial autocorrelation.

(10) The pick of the 12 paper that are investigated seemed a bit random. Certainly, there are more famous papers...

We wanted to be as systematic as possible in our choice of papers to reanalyse, so we focused papers that (1) were included in our review, (2) had available data for reanalysis, and (3) we could replicate the analysis for. Some of the more famous papers in our review were not included in the reanalysis because they did not make their data available or we could not replicate their main findings.

Enrico Spolaore, Romain Wacziarg (2009). “The Diffusion of Development”, The Quarterly Journal of Economics, Volume 124, Issue 2, May 2009, Pages 469–529, <https://doi.org/10.1162/qjec.2009.124.2.469>

Schulz, Jonathan (2022). “Kin Networks and Institutional Development”, Economic Journal

Fabrizio Colella, , Rafael Lalive, Seyhun Orcan Sakalli, and Mathias Thoenig (2019). “Inference with Arbitrary Clustering” Working paper

Joachim Voth (2021). “Persistence: Myth and Mystery”, In Bisin and Federico, Handbook of Historical Economics

Reviewer 4

This is an important paper. It presents evidence that nation states are not independent datapoints for social science research; that many published studies do not appropriately control for this

non-independence; and that not controlling for this issue increases the false positive rate. While the problem of non-independence of nation states has been known for some time, in at least some social science disciplines, this paper provides a very clear, rigorous appraisal of this problem, adding new evidence that not controlling for the problem likely biases the academic literature. I recommend publication, and have only minor comments for the authors to consider.

The methods appear to be sound. This is quite a complex paper in that the authors present a series of analyses: (1) testing whether nation states can be considered independent in social science research; (2) investigating the published literature to determine whether controls for non independence are commonly used; (3) performing a simulation analysis showing that not controlling for non independence increases the false positive rate, highlighting a Bayesian method which controls for non independence and reduces the false positive rate down to reasonable levels, and showing that commonly used frequentist controls for non independence don't typically reduce the false positive rate down to acceptable levels; (4) re-running analyses for 12 published studies to show that the findings cannot be replicating in half of the studies when using methods which appropriately control for non independence. However, the analyses are clearly justified and explained, the methods section is well described, and the narrative flow of the paper is good.

We thank this reviewer for their positive comments on our manuscript.

My only comment about the methods is a request for more clarity in the final section of the paper – the re-analysis of published studies. The authors give limited information on how they chose 12 studies from the original 100, saying only that they choose studies where the data were available and they could replicate the original finding. Does this mean that for the other 88 studies chosen for earlier analysis, either the data could not be found or the original finding could not be replicated? If so, that seems noteworthy and could be mentioned in the paper. If instead, the authors didn't systematically try to replicate all 100 studies, but stopped when they had found 12 studies with available data where they could replicate the original finding, that could also be made clear. Were any other exclusion criteria used? For example, studies that did not find positive associations might have been excluded? This might also be useful for the reader to know.

We agree that the inclusion criteria were unclear in the previous draft. In fact, 47 of the 100 papers had at least some available data (we now report this in the paper). But to get these data into a workable state, many analyses required complicated matching of external datasets, to the point where it became infeasible to reanalyse all 47 datasets. We opted to instead choose an even selection of papers from each of the economic development and cultural values reviews, ideally well-known papers. We stopped when we successfully replicated the initial specifications for 12 of these previous statistically significant correlations, finding similar effect sizes to those reported in the original papers. Importantly, we pre-registered this set of analyses before running any further controls. We now make these details clearer in the main text.

I have a couple of small suggestions for additional discussion points:

1. It might be helpful to include some discussion of whether the 50% non-replication rate the authors found for published studies might approximately hold across all published studies which have used cross-national analysis. My guess might be that an even higher proportion of published studies would not replicate, given that the studies which the authors were able to replicate might be more rigorous than other studies – assuming that studies in which data could be found and the original result replicated were higher quality than others. But this is perhaps getting into speculation the

authors would not find helpful. This is just a soft suggestion for some additional discussion the authors might want to include.

We agree that our reanalyses point to a potential bias in the published literature. We have expanded our discussion of this point in our Discussion section.

2. As another suggested discussion point – can the authors provide any hope for the frequentists among us, that there might be frequentist controls that could reduce the false positive rate down to reasonable levels? Their conclusion is that Bayesian methods work better than frequentist for this problem – is this because there is something about Bayesian methods which is superior for the particular problem described here? If so, perhaps the authors could explain why. (I'm not recommending the authors include an essay on the merits of Bayesian versus frequentist stats here, just some insight into why Bayesian methods might have performed better in this particular instance).

We are not aware of frequentist versions of the Bayesian Gaussian Process or correlated random effects models that we conduct in this paper. However, our new simulation shows that controlling for the mean of the predictor variable within a 2000km radius (Schulz et al. 2022) is a frequentist method that effectively reduces false positives under spatial non-independence, with the caveat that this method has lower statistical power than the Bayesian approaches and does not adequately deal with cultural non-independence. We now mention this method in the Discussion section.

It is worth noting that there is nothing superior about Bayesian methods *per se*. The Bayesian approach is simply better suited to fit models with complex covarying random effects structures. If frequentist versions of these models existed, they would likely perform equally well. We now briefly explicate this in the Discussion.

To aid readers unfamiliar with the Bayesian approaches, in the paper we now link to a helpful blog tutorial that explains how to run these models in R (<https://scottclaessens.github.io/blog/2022/crossnational/>).

My final comment is not one which is relevant to whether this paper should be published or not, but I wonder do the authors intend to (or have they already) contacted the authors of analyses whose significant findings they were not able to replicate? Or the editors of journals which published those papers? The analyses presented here suggest that those authors or editors should consider correcting or retracting the papers with non-replicable findings.

We have not contacted the authors or journal editors of these previous papers, nor do we intend to. While we understand the want to update the scientific record, we do not believe that the results of our reanalyses warrant corrections or retractions. Instead, we believe that science is a slow, incremental process, where different causal models and regression specifications should be openly discussed, debated, and refined. Our hope is that, with our open data and code, the original authors should be able to inspect our analytic decisions and rebut any that they disagree with. Keeping a record of these debates in the literature is the sign of a healthy constructive science.

Reviewer 5

The main finding of this work is that existing (economic and psychological) cross-national studies is unable to control for spatial and linguistic (“cultural”) non-independence between nations.

As a socio-cultural anthropologist, I am overall sympathetic toward any attempt to highlight and systematically investigate (via the use of qualitative or quantitative methods) the potential importance of often overlooked features and variables in economists’ and psychologists’ comparisons between different social and cultural units or scales, including nations. But for the same reason, I am also skeptical towards any attempt (whether deliberate or as a result of theoretical sleight-of-hand) to reduce the differences and similarities between two or more such units to a mere question of spatial and/or cultural proximity. In particular, I find the exclusion of political-economic structures, traditions and practices (e.g. communist/post-communist versus liberal/neoliberal) to constitute a potential major problem in terms of the both the construct (what is being studied) and measurement (how it is being studied) validity of the present study. Consider, for example, the case of the former so-called Second World (the Soviet Union and its economic and political dependences around the world, including not just Eastern Europe but also Cuba and Vietnam, etc.). As been demonstrated by a number of prominent sociologists (e.g. Dunn, Burawoy, Ledeneva) and anthropologists (e.g. Yurchak, Verdery, Humphrey), routines and values embodied and institutionalized in the course of 2-3 generations of state socialism lived on (and still today live on) as a distinct post-socialist culture, which clearly needs to be taken into account when designing, operationalizing and (meta) data processing a study like the present one.

We do not deny that histories of political-economic structures, traditions, and practices, such as state socialism, have had a marked causal impact on the modern-day economic development and cultural values of nations around the world. Indeed, for this very reason, cross-national studies in economics and psychology often do include controls like post-communist status in their statistical models. However, our goal here was not to construct complete causal models of economic development and cultural values around the world, but rather to highlight how two specific forms of non-independence between nations, spatial and cultural, can lead to biases in cross-national statistical inference. In the Discussion section, we now state that controls for non-independence should be included alongside any other necessary control variables (e.g., post-communist status), as justified by a causal model specific to the analytic problem at hand.

It is also worth noting that these forms of non-independence are antecedent to (not mutually exclusive from) the expansion of the kinds of political-economic structures the reviewer mentions. As a result, any investigation into the associations between national-level outcomes and political-economic structures also needs to control for spatial and cultural non-independence in the data — for example, when evaluating the impact of communism on, say, economic development, one needs to control for the fact that many communist nations are clustered in Eastern Europe and have a Slavic cultural heritage, and so may share many features besides their history of communism.

As already noted, I find the questions posed by and the conclusions from this study to be of potentially significant interest to scholars and students from across the social sciences, even if the nature of the approach as well as the data and the methods are such that it will mostly appeal to readers trained in quantitative, empirical/experimental traditions. On a related point, while the study seems to be well-grounded in and positioned with respect to the state of the start within behavioral science and evolutionary anthropology, it is quite thin when it comes to engaging with relevant work

from sociology and, especially, social and cultural anthropology. To be sure, the authors cannot be expected to be in possession of any expert knowledge or overview of these neighboring fields, but given that they explicitly mention “anthropology” as a source of identity and inspiration (page 19, line 318), one would expect them to reference at least a few major works from socio-cultural anthropology, which also engage with “non-independence between nations”, albeit from very different theoretical and methodological perspectives and approaches (think, for example, of Ferguson’s work on Africa, and Luhrmann’s comparative work on Christian belief and prayer).

We thank the reviewer for these references to related work in social and cultural anthropology. Indeed, Ferguson (1997) explores the history of anthropological views on the development of nations, describing how theories of unilineal independent development were supplanted in the 1970s by theories that emphasised the development of a globally interconnected “world system”. For example, “dependency theory” attributes variation in development around the world to common histories of imperialism and economic exploitation. We now refer to this work in our introduction section.

In terms of data quality, clarity of the presentation and reproducibility, I have little or nothing to say – but then again, this could be due to the fact that I am quite unfamiliar with this genre (statistically driven meta-study) and field (behavioral science/evolutionary anthropology) of research. When respect to the question of validity of the approach, however, I have several questions and reservations. For one thing, I am (as already pointed out) quite concerned about the lack of political-economic constructs and variables in the study. Perhaps this point can best be illustrated with reference to Supplementary Figure S1: while I certainly agree with the authors that “geographic (G) and linguistic (L) relationships” need to be taken into consideration in cross-national research, why does the same not go for “political” (P) and “economic” (E) – or better still, political economic (PE) ones? In fact, the authors do mention so-called “institutional causes” (page 41) as a potential variable, but they only do so in passing and features pertaining to political-economic structures and traditions (e.g., collective land rights, non-liberal forms of governance, etc.) do not seem to play any role in the authors’ operationalization neither G or L.

As previously mentioned, our goal with the study (and with Supplementary Figure S1) was not to capture and account for all relevant causes of economic development and cultural values. Instead, we wanted to focus on the sources of bias arising from statistical non-independence specifically. Certainly, *some* of the global variation in important political-economic variables will be captured by controls for spatial and cultural non-independence, but by no means all. For this reason, we now highlight the importance of including controls for non-independence alongside any other necessary control variables (e.g., post-communist status), as justified by a causal model specific to the analytic problem at hand.

Indeed, and this is my second reservation, the author mostly seem to reduce cultural variation to linguistic ditto, and on the occasions when they do engage with extra-linguistic aspects of cultural life, they turn these into just two measures, “traditional vs. secular values” and “survival vs. self-expression values” (p. 21, line 364-370). It is true that the latter operationalization is justified with reference “to previous research” (ibid), but tellingly the research in question only amounts to a single article published in *American Sociological Review*! To repeat a previous point of mine, no-one could reasonably expect the authors do be able – or indeed willing – to engage in detail with the extremely extensive socio-cultural (as well as cultural sociological) literature pertaining to the complex, dynamic and interconnected ways in which cultural patterns, processed and dynamics are embedded within and influenced by political-economic structures and dynamics. However, at a minimum they should at least mention the existence of this literature and some of its main

conclusions and insights, even if the purpose of doing so is to refute its relevance for the present study, or to criticize the overarching assumptions of such “holistic” and “contextual” approaches.

We value the rich, nuanced descriptions of cultural variation and change provided by ethnographies and case studies in cultural anthropology and sociology. As noted by the reviewer, this literature broadly acknowledges the myriad ways in which human populations around the world are intimately connected via, for example, shared histories of colonialism and the global spread of capitalist modes of production (Trouillot, 2016; Wolf, 2010). In our Discussion section, we now explain how our work complements these views, and suggest that future research explore other important connections between nations (e.g. political-economic dynamics).

Regarding our particular quantitative measures of non-independence, we agree that there is more to cultural history than linguistic genealogies, such as common descent in material culture, religious traditions, and kinship terms. Likewise, there is more to geography than raw distances between nations, such as whole ecosystems and biological and social worlds traversed across distances. But this does not mean that these measures are not extremely useful for understanding and accounting for statistical dependencies between nations. Linguistic and geographic distances are convenient proxies for more complex processes — they are easy to measure and use, process-based, and general. Moreover, our first analysis and other work (e.g. Matthews et al. 2016) shows that these measures *do* capture variation in a variety of national-level outcomes.

Regarding the choice of cultural variables in our first analysis, we agree that there is more to cultural life than traditional vs. secular values and survival vs. self-expression values. We chose these variables as examples because they have been previously identified as two major axes of cultural variation and are widely used and cited (Bloom & Arikan, 2013; Dobewall & Rudnev, 2014; Inglehart, 2006, 2007; Inglehart & Baker, 2000). We are not claiming that culture can be fully captured by these two variables, nor are we claiming that geography and language always explain a large amount of variation across all human culture. Rather, our point was that these factors are potentially important for commonly studied measures of culture in the literature. We hope that we have dealt with this criticism somewhat by incorporating further cultural variables into our first analysis: Michele Gelfand’s cultural tightness and Geert Hofstede’s individualism.

Given my background in socio-cultural anthropology, I do not have the technical competences to adequately evaluate any specific questions pertaining to the paper’s use of statistics and treatment of uncertainties. However, based on my experience from collaborating with computational social scientists, I am wondering whether one potential source of model bias come from the use of statistical R packages such as *brms* and *conleyreg*, which (as the authors also note on Page 37, line695-697) come with certain inbuilt and thus potentially black boxed assumptions.

While many statistical packages can be seen as black boxes, the procedures and assumptions underlying these two open-source R packages are not hidden from the user. The *brms* R package is a front-end wrapper for the Stan programming language. Stan code can be extracted from any model fitted with *brms*, which explicitly outlines all model assumptions, including prior specifications. Similarly, all R code underlying the *conleyreg* R package is publicly available online (<https://github.com/cran/conleyreg>).

Our simulation study explicitly measured the amount of bias in these different open-source methods (i.e. false positive rates). In the simulation, we knew the true relationship between our variables, and found that the Bayesian random effects approach was able to overcome the biases that other modelling approaches suffered from. In a sense, then, models from the *conleyreg* package are biased, but this bias arises from open-source procedures and not from any hidden “black-box” assumptions. A strength of our study is that we measure and report the extent of this bias.

In summary, I think this is an interesting meta-study, whose results (and general message regarding the weaknesses with cross-national studies) are likely to be of interest to a wide range of scholars and students from the social, economic and behavioral sciences. However, further work needs to be done on the study’s theoretical constructs, as well as the operationalization of these variables into specific measures. In particular, the authors need to consider the importance of non-spatial and non-linguistic relationships (e.g. shared political history and culture), something which might best be accomplished by delving deeper into the relevant sociological and socio-cultural anthropology literature.

We thank the reviewer for their insightful comments. We believe that the changes we have made in response to these criticisms – including further cultural variables in our first analysis, discussing how our results complement theoretical approaches in socio-cultural anthropology, suggesting that future research explore additional political-economic connections between nations – have resulted in an improved manuscript.

Reviewer 6

In their manuscript “The non-independence of nations and why it matters”, the authors make a single point: Most cross-national analyses do not sufficiently control for the statistical covariance among nations that arises because of spatial and/or cultural proximity. They make this point extremely well. I find the paper very clear, well written and (especially for a methods paper) really entertaining. All re-analyses were preregistered and the authors provide the full data and analysis code on GitHub (when trying to clone the repo, I encountered some problems with the “Integrated_values_surveys_1981-2021.sav” file, maybe the authors want to check if there is indeed some problem).

We thank this reviewer for their positive comments on our manuscript. We are also grateful to the reviewer for taking a look at the code and attempting to clone the repository. It is our aim that the code be as reproducible as possible. Regarding the cloning issue, it seems that there was an error with Git LFS that stores large files. After fixing the error, it should now be possible to clone the repository, provided that the user has Git LFS installed (<https://git-lfs.github.com/>). We have updated the README to reflect this.

As requested by the editor, I will focus my comments mainly on the Bayesian modeling methods used here. The analyses are rigorous, clearly described and overall appropriate for the research question. I also appreciate the full mathematical model details in the supplementary information which greatly helped me evaluate their models. However, I also have some concerns and comments.

First, the authors seem to suggest that their phylogenetic/Gaussian process regression approach generally controls for the non-independence of nations. As described in Fig. S1, the problem this approach is addressing is that unobserved variables U influence both X and Y and thus confound the

causal effect. Their approach tries to model the covariation among populations that arises from such unobserved confounds. However, this method is no panacea for causal inference and also makes strong assumptions about the nature of confounding and our ability to measure cultural proximity. Specifically, the method only sufficiently controls for U if geographic G and linguistic relationships L (as defined by the user) are indeed the only causes of U. In general, such methods will only work if researchers can find good proxies for U and the success will depend on how closely such proxies correspond to U, with spatial and linguistic proximity only being two candidate variables. In a different context, variables other than G and L might be more appropriate to model U, the general point being that researchers should explicitly model the co-variance among nations to take account of their non-independence. Additionally, the models assume that linguistic similarity accurately reflects the covariance of those cultural traits that influence the unobserved nation-level variables U. Of course, strong assumptions are always necessary for causal inference in observational settings (Rohrer, 2018), so this is not a problem per se, the authors should just make those assumptions more transparent to not overexaggerate the power of this method and make readers aware of its inferential limits.

There are several points here. First, we agree that in certain contexts there may be other important sources of non-independence at work aside from geographic and linguistic proximity. For example, nations are also connected by a variety of modern socio-economic networks (e.g., flight networks and social media connections) and shared political histories which may also influence U. We refer to these additional sources of non-independence as a direction for future research in the Discussion section.

Second, we acknowledge that the example causal model in Supplementary Figure S1 does not capture other causes of economic development and cultural values, such as the political-economic variables highlighted by reviewer #5. Our goal with this toy model was to describe the general modelling approach, rather than to account for all relevant causal influences on economic development and cultural values. In the Discussion section, we state that controls for non-independence should be included alongside any other necessary control variables, as justified by a causal model specific to the analytic problem at hand.

Third, we admit that linguistic similarity is but one proxy for cultural relatedness. Many cultural traits might be better captured by other metrics of cultural relatedness, such as religious similarity or cultural FST values (Muthukrishna et al., 2020). We now mention such alternative proxies in our Discussion section.

Next, I find the way the authors use Gaussian processes to model spatial proximity very sensible and, thus, do not understand why they seem to use a much more rigid approach for cultural similarity where they assume a fixed, linear relationship between phylogenetic distance and covariance. As rightly pointed out by the authors, their approach is justified if “cultural traits evolve via Brownian motion along a language phylogeny”. However, most cultural traits do not evolve neutrally, and also evolve at different rates in different parts of the tree (assuming that linguistic phylogenies describe the evolution of relevant cultural traits at all...). The authors might consider using Gaussian processes for this analysis as well to estimate the covariance function and also allow for non-linear relationships between phylogenetic distance and covariance (see McElreath, 2020, chapter 14.5.2.).

We are sympathetic to this point, however we would prefer to stick to the Brownian motion assumption in this study, for several reasons. First, in previous attempts to model both geographic and linguistic covariance simultaneously with two separate Gaussian Processes, the first author has often run into issues of model non-convergence. Second, since linguistic

distance does not have a simple coordinate system like longitude and latitude, a Gaussian Process for this distance matrix would require the models to be specified in the *rethinking* R package or in raw Stan code (see discussion here). This isn't a problem in and of itself, but the *brms* package was useful to use as it allowed us to additionally fit *approximate* Gaussian Processes, which often became necessary in our reanalysis section to avoid non-convergence issues. Third, we also wanted to showcase different ways that researchers can allow nations to covary, and the Brownian motion approach is one that is frequently used in the phylogenetic and cultural phylogenetic literature (see here). That said, we have acknowledged the limitations of the Brownian motion assumption when discussing the model in the supplementary materials.

Figure 6 shows the effect of the estimated spatial/cultural phylogenetic signal on the reductions in effect size in their re-analyses. Both variables are model estimates not data, but the analysis seems to ignore this uncertainty and only include the posterior medians instead of the full posteriors. This is a problem as there is usually quite some uncertainty in Gaussian-process control parameters which is also evident from Fig. S5. I would suggest the authors to repeat this analysis with the full posteriors instead of only posterior means to accurately reflect the full uncertainty.

This is an excellent point. We have reconducted these models iterating over the posterior reductions in effect sizes *and* spatial / cultural phylogenetic signal estimates. While the slopes in both of these models are negative, the 95% credible intervals are wide and include zero, likely due to the posterior uncertainty and the sample size of only 12 studies. Given this uncertainty, we have moved these results to the supplementary materials.

More broadly, the problem of non-independence between nations is just one potential pitfall in cross-national research and the authors might want to discuss links to wider concerns about causal inference in cross-national/cross-cultural research. Valid comparisons and inference in a cross-national setting requires that researchers make assumptions about the precise mechanisms by which populations might differ (in this example: distance and linguistic similarity are the only mechanisms that generate co-variance between X and Y) and how the data are generated, and design tailored estimation strategies based on those causal assumptions (see Deffner et al., 2022). As one example, Deffner et al. (2022) discuss the "proxy control" approach using phylogenetic distance that is used in this ms (Fig. 7b), but also discuss fundamental issues of measurement, description and generalization.

We have now expanded our discussion of causal inference towards the end of the manuscript. Referring to Deffner et al. (2022) and others, we clarify that our causal model in Supplementary Figure S1 is only an example, and that each cross-national study should clearly outline its particular causal assumptions to justify its own estimation strategy.

Lastly, in several places (e.g., lines 209, 218, 222), the authors present their "Bayesian" approach as an alternative to "frequentist methods" which do not sufficiently control for non-independence. However, the question here is not really about Bayesian or frequentist estimation, but about the random-effects covariance structure among nations. Therefore, also Bayesian models which do not model how this covariance arises from relevant causes will be inappropriate. Related to that, in the abstract, the authors write that they show that half of results "are no longer significant after controlling for non-independence", which is misleading as they are not using classic significance testing and only some model estimates substantially change compared to the original values even if some arbitrary HPDI now overlaps 0.

We have now made clearer in the manuscript that the methods we use are superior not because they are Bayesian *per se*, but because they allow nation random intercepts to covary with one another in a structured way. We have also amended the wording about “significant” results (sometimes it is easy to slip back into significance testing language!).

Cited literature

Deffner, D., Rohrer, J. M., & McElreath, R. (2022). A causal framework for cross-cultural generalizability. PsyArxiv. <https://doi.org/10.31234/osf.io/fqukp>.

McElreath, R. (2020). *Statistical rethinking: A Bayesian course with examples in R and Stan*. Chapman and Hall/CRC.

Rohrer, J. M. (2018). Thinking clearly about correlations and causation: Graphical causal models for observational data. *Advances in methods and practices in psychological science*, 1(1), 27-42.

References

Bloom, P. B. N., & Arikan, G. (2013). Religion and support for democracy: A cross-national test of the mediating mechanisms. *British Journal of Political Science*, 43(2), 375-397.

Bromham, L., Hua, X., Cardillo, M., Schneemann, H., & Greenhill, S. J. (2018). Parasites and politics: why cross-cultural studies must control for relatedness, proximity and covariation. *Royal Society Open Science*, 5(8), 181100.

Dobewall, H., & Rudnev, M. (2014). Common and unique features of Schwartz's and Inglehart's value theories at the country and individual levels. *Cross-Cultural Research*, 48(1), 45-77.

Ferguson, J. (1997). Anthropology and its evil twin. *International development and the social sciences: Essays on the history and politics of knowledge*, 150-175.

Guiso, L., Sapienza, P., & Zingales, L. (2009). Cultural biases in economic exchange? *The Quarterly Journal of Economics*, 124(3), 1095-1131.

Handley, L. J. L., Manica, A., Goudet, J., & Balloux, F. (2007). Going the distance: human population genetics in a clinal world. *TRENDS in Genetics*, 23(9), 432-439.

Inglehart, R. (2006). Mapping global values. *Comparative Sociology*, 5(2-3), 115-136.

Inglehart, R. (2007). Postmaterialist values and the shift from survival to self-expression values. *The Oxford Handbook of Political Behavior*, 223-239.

Inglehart, R., & Baker, W. E. (2000). Modernization, cultural change, and the persistence of traditional values. *American Sociological Review*, 65(1), 19-51.

Kelly, M. (2019). The standard errors of persistence. *CEPR Discussion Paper No. DP13783*, Available at SSRN: <https://ssrn.com/abstract=3401870>

Kelly, M. (2020). Understanding persistence. *CEPR Discussion Paper No. DP15246*, Available at SSRN: <https://ssrn.com/abstract=3688200>

Matthews, L. J., Passmore, S., Richard, P. M., Gray, R. D., & Atkinson, Q. D. (2016). Shared cultural history as a predictor of political and economic changes among nation states. *PLOS ONE*, *11*(4), e0152979.

Muthukrishna, M., Bell, A. V., Henrich, J., Curtin, C. M., Gedranovich, A., McInerney, J., & Thue, B. (2020). Beyond Western, Educated, Industrial, Rich, and Democratic (WEIRD) psychology: Measuring and mapping scales of cultural and psychological distance. *Psychological Science*, *31*(6), 678-701.

Novembre, J., Johnson, T., Bryc, K., Kutalik, Z., Boyko, A. R., Auton, A., ... & Bustamante, C. D. (2008). Genes mirror geography within Europe. *Nature*, *456*(7218), 98-101.

Nunn, N. (2020). The historical roots of economic development. *Science*, *367*(6485), eaaz9986.

Schulz, J. F., Bahrami-Rad, D., Beauchamp, J. P., & Henrich, J. (2019). The Church, intensive kinship, and global psychological variation. *Science*, *366*(6466), eaau5141.

Schulz, J. F. (2022). Kin networks and institutional development. *The Economic Journal*, *132*(647), 2578-2613.

Spolaore, E., & Wacziarg, R. (2009). The diffusion of development. *The Quarterly Journal of Economics*, *124*(2), 469-529.

Spolaore, E., & Wacziarg, R. (2013). How deep are the roots of economic development? *Journal of Economic Literature*, *51*(2), 325-69.

Spolaore, E., & Wacziarg, R. (2016). War and relatedness. *Review of Economics and Statistics*, *98*(5), 925-939.

Trouillot, M. (2016). *Global Transformations: Anthropology and the Modern World*. Springer.

Villemereuil, P. D., & Nakagawa, S. (2014). General quantitative genetic methods for comparative biology. In *Modern phylogenetic comparative methods and their application in evolutionary biology* (pp. 287-303). Springer, Berlin, Heidelberg.

Wolf, E. R. (2010). *Europe and the People without History*. University of California Press.

REVIEWER COMMENTS

Reviewer #1 (Remarks to the Author):

The authors have done a great job addressing my critiques. I think the paper offers a compelling case for new methods of controlling for cultural non-independence, and also provides tools for researchers to implement these controls. I can imagine this paper prompting a new standard in cross-cultural research.

Reviewer #3 (Remarks to the Author):

Thank you very much for this revised draft. I think this is a great research. Getting guidance on how to approach the important topic of spatial and cultural autocorrelation in a very practical manner is very helpful. I very much appreciate the new analyses.

I have one concern on how you portray the existing debate on spatial autocorrelation in the economic discipline on persistence. I don't think it is helpful both for your paper (that gives great insights) nor for the scientific discourse "to pick sides" and then propagate an unbalanced narrative. The reason I suggested to check out Voth (2021) is to get a more balanced view on Kelly (2019). Voth (2021) makes three points that can easily be verified as objectively true: (i) spatial autocorrelations is not a specific problem to the persistence literature; (ii) Kelly's selective choice of the specifications; (iii) the importance of other specification that form part of the paper (i.e., the first-specification in a paper is usually a simple cross-country relationship while the heavy lifting in econ papers often come from other specifications at a lower level of aggregation).

For example, in your paper you show that the cross-country relation in Alesina's et al. (2013) plough paper may not be significant when addressing spatial and cultural autocorrelation. In the econ community it is very clear that cross-country regressions should be taken with a grain of salt since they are not able to address all potential confounding variables. Consequently, the Alesina et al (2013) paper has a host of other analyses and datasets including individual-level and a second-generation immigrant analysis. Does spatial autocorrelation pose a problem for this other analyses? We don't know from your paper but my strong prior is that for individual-level analysis which often contains many more individually varying observations and country or region fixed effects that this is quite unlikely.

So, yes, spatial and cultural autocorrelation is important (you demonstrate this in your paper at the country level). Yet, the evidence you or Kelly (2019) presents at the country level can hardly be taken to reject specific papers or a whole literature. Thus, why not acknowledge this (i.e., since you only pick one specification at the nation level and many papers have a host of other analysis your analyses do not imply that the key hypothesis of the papers break down). The word "ilk" is often used in a disparaging way. Is it your intention to signal that you do not like this literature even though spatial autocorrelation is not a problem that is specific to this literature?

Taken together, I think you have an important paper and there is no need to give an unbalanced or misleading view of an existing literature just to make your paper appear more important.

Small quibbles about your replies:

(1) Nation vs country: I do not agree that "Unlike (...) "country" (...), the label "nation" does not presuppose any particular form of political governance." I don't think that people associate a specific political governance with country while this is not the case for "nation". I agree when you write that nation "emphasises the deep cultural and historical ties between human populations that create global non-independencies". Yet, this is precisely why I believe country is the better term. Most countries in the world do not have populations that share deep cultural and historical ties. They are ethnically

heterogeneous – particularly in Africa. Thus, while an analysis of nations (as a homogenous unit) would be desirable, the empirical reality is that you cannot. You rather analyze geographic units that are delineated by administrative borders.

(2) I don't agree with your characterization of language vs genetic distance. (Please do not view this as something you should incorporate in your paper now – totally fine to only discuss linguistic distance). I think genetic distance is the better measure because it is better able to capture cultural distance. The reason is that linguistic distance rests on phylogenetic trees. Nodes in these trees (or hierarchies) are not necessarily able to capture the distance well. That is, moving up the hierarchy by one node might reflect a very different cultural distance depending on the specific language (or dialect). That is, nodes reflect categories and not a continuous distance. I don't see how the points you list in favor of linguistic distance speak against genetic distance. First, just like genetic distance, linguistic distance will be highly correlated with geography and it is not clear why this correlation is a good criteria to choose one over the other measure. Second, culture is a fluid and just like genetic distance consistent with multiple underlying process. For example, a huge migration inflow will alter a countries genetic distance but also its cultural distance because immigrants bring both – genetic and cultural endowments. In case these migrants switch their language while keeping many of their other cultural traits genetic distance would be better able to catch this compared to linguistic distance. This consideration makes it also obvious that it is by no means clear why language should be more clearly tied to "specific underlying processes (i.e., deep cultural inheritance)." and it is not clear why this should be a desirable property of a measure of cultural distance.

These "quibbles" should not distract from the fact you wrote a great paper. Congrats.

Reviewer #4 (Remarks to the Author):

I thank the authors for responding to my comments, and am now satisfied that this manuscript will make a very fine contribution to the literature.

Reviewer #5 (Remarks to the Author):

I have had a good look at both the revised manuscript and the authors' comments, which I very much appreciated.

I am happy to support the publication of this paper in its current, revised format

Reviewer #6 (Remarks to the Author):

I would like to thank the authors for this comprehensive and thorough revision of their manuscript. All of my comments have been addressed and I would therefore like to congratulate the authors for this excellent paper. I've also been able to clone the repo now and run (some of) the analyses.

Response to Reviewers

Reviewer 3

Thank you very much for this revised draft. I think this is a great research. Getting guidance on how to approach the important topic of spatial and cultural autocorrelation in a very practical manner is very helpful. I very much appreciate the new analyses.

We thank this reviewer for their positive comments on our revised manuscript. We also believe that the paper has been much improved by the revision.

I have one concern on how you portray the existing debate on spatial autocorrelation in the economic discipline on persistence. I don't think it is helpful both for your paper (that gives great insights) nor for the scientific discourse "to pick sides" and then propagate an unbalanced narrative. The reason I suggested to check out Voth (2021) is to get a more balanced view on Kelly (2019). Voth (2021) makes three points that can easily be verified as objectively true: (i) spatial autocorrelations is not a specific problem to the persistence literature; (ii) Kelly's selective choice of the specifications; (iii) the importance of other specification that form part of the paper (i.e., the first-specification in a paper is usually a simple cross-country relationship while the heavy lifting in econ papers often come from other specifications at a lower level of aggregation).

For example, in your paper you show that the cross-country relation in Alesina's et al. (2013) plough paper may not be significant when addressing spatial and cultural autocorrelation. In the econ community it is very clear that cross-country regressions should be taken with a grain of salt since they are not able to address all potential confounding variables. Consequently, the Alesina et al (2013) paper has a host of other analyses and datasets including individual-level and a second-generation immigrant analysis. Does spatial autocorrelation pose a problem for this other analyses? We don't know from your paper but my strong prior is that for individual-level analysis which often contains many more individually varying observations and country or region fixed effects that this is quite unlikely.

So, yes, spatial and cultural autocorrelation is important (you demonstrate this in your paper at the country level). Yet, the evidence you or Kelly (2019) presents at the country level can hardly be taken to reject specific papers or a whole literature. Thus, why not acknowledge this (i.e., since you only pick one specification at the nation level and many papers have a host of other analysis your analyses do not imply that the key hypothesis of the papers break down). The word "ilk" is often used in a disparaging way. Is it your intention to signal that you do not like this literature even though spatial autocorrelation is not a problem that is specific to this literature?

Taken together, I think you have an important paper and there is no need to give an unbalanced or misleading view of an existing literature just to make your paper appear more important.

We concede that our discussion of the persistence literature in the revised manuscript did not fully capture the existing debate. As in Kelly (2019), our country-level reanalyses focus only on the initial regression specifications presented in papers. As a result, we are unable to say anything more about the robustness of subsequent regression specifications or individual-level analyses, and are therefore unable to outright reject the claims made in these papers. Any comprehensive attempt to challenge these claims would require more detailed sets of reanalyses justified by causal models for each specific claim, which is beyond the scope of this manuscript. We have amended the Discussion section to be clearer about what we can and cannot claim from our results.

That said, we believe that the results of our reanalyses *do* suggest that the conclusions drawn from these papers should, at the very least, be subjected to further scrutiny. As we mention in our Discussion section, we see a key strength of the statistical tradition in economics to be the gradual inclusion of control variables to initial bivariate regressions in order to make causal claims. In our paper, we have done exactly this by including controls for spatial and/or cultural autocorrelation to the initial regression specifications from papers. If the results from the initial specifications are not robust to these controls, this raises the possibility that autocorrelation is generating a spurious relationship. The same patterns of non-independence might also be affecting other regression specifications, even individual-level analyses of cross-national data. We now mention in our Discussion section that future work should explore such possibilities in more detailed reanalysis studies.

The goal with our reanalyses was never to target individual papers. The twelve analyses we focused on were just the cross-national correlations from our review that we were able to replicate with available data. Rather, our goal was to highlight a more general inferential problem that could potentially apply to the 56% of economic development papers and 92% of cultural values papers that ignore the issue of autocorrelation completely. We now state this clearly in the Discussion section. We have also amended the wording to avoid disparaging the persistence literature, which was not our intention.

Small quibbles about your replies:

(1) Nation vs country: I do not agree that “Unlike (...) “country” (...), the label “nation” does not presuppose any particular form of political governance.” I don’t think that people associate a specific political governance with country while this is not the case for “nation”. I agree when you write that nation “emphasises the deep cultural and historical ties between human populations that create global non-independencies”. Yet, this is precisely why I believe country is the better term. Most countries in the world do not have populations that share deep cultural and historical ties. They are ethnically heterogeneous – particularly in Africa. Thus, while an analysis of nations (as a homogenous unit) would be desirable, the empirical reality is that you cannot. You rather analyze geographic units that are delineated by administrative borders.

This is an interesting point. Of course, we agree that modern nations are often ethnically heterogeneous – indeed, our approach to quantifying cultural distance incorporates this fact. However, this is not incompatible with standard modern use of the term “nation”. The Oxford English Dictionary defines a nation as “*a large aggregate of communities and individuals united by factors such as common descent, language, culture, history, or occupation of the same territory, so as to form a distinct people. Now also: such a people forming a political state; a political state.*” As this definition alludes to, the term “nation” has historically emphasised the role of shared linguistic and cultural ancestry, which is the subject of our paper, but in recent usage its meaning has widened to include any political state occupying a given territory. In the scientific literature, the terms “country” and “nation” are therefore often used interchangeably, as is demonstrated by studies we cite (e.g. Gelfand et al. 2011 vs. Beck et al. 2005). And notably, “nation” is now used by reputable organisations such as the United Nations to refer to any modern state. We therefore strongly favour using the term “nation” in our paper, which is a legitimate term for any modern state but harks back to the older usage of the term as alluding to shared cultural ancestry.

(2) I don’t agree with your characterization of language vs genetic distance. (Please do not view this as something you should incorporate in your paper now – totally fine to only discuss linguistic

distance). I think genetic distance is the better measure because it is better able to capture cultural distance. The reason is that linguistic distance rests on phylogenetic trees. Nodes in these trees (or hierarchies) are not necessarily able to capture the distance well. That is, moving up the hierarchy by one node might reflect a very different cultural distance depending on the specific language (or dialect). That is, nodes reflect categories and not a continuous distance. I don't see how the points you list in favor of linguistic distance speak against genetic distance. First, just like genetic distance, linguistic distance will be highly correlated with geography and it is not clear why this correlation is a good criteria to choose one over the other measure. Second, culture is a fluid and just like genetic distance consistent with multiple underlying process. For example, a huge migration inflow will alter a countries genetic distance but also its cultural distance because immigrants bring both – genetic and cultural endowments. In case these migrants switch their language while keeping many of their other cultural traits genetic distance would be better able to catch this compared to linguistic distance. This consideration makes it also obvious that it is by no means clear why language should be more clearly tied to “specific underlying processes (i.e., deep cultural inheritance).” and it is not clear why this should be a desirable property of a measure of cultural distance.

This is another interesting debate. We see merit in both approaches, and we have made minor edits to our Discussion to better reflect this (see lines 340-348). However, we used linguistic distance in our study because we believe it offers some advantages over genetic distance as a measure of cultural distance. While it is beyond the scope of this paper to fully address this debate, we wanted to at least justify our choice of linguistic distance here.

First, it is important to clarify that phylogenetic trees of languages can and often do track continuous (rather than simply categorical) distances between taxa. Patristic distances from linguistic phylogenies based on time or amount of linguistic change can therefore also be continuous measures. In our analysis, we use diversification events as a proxy for the amount of change, but since we also incorporate information on the percentage of speakers of each language in each nation, even this measure is effectively continuous.

Second, while linguistic distance is sometimes positively correlated with geographic distance, the association is not as reliably positive as the association between genetic distance and geographic distance (Chen et al., 2012; Table 1). This is an issue when using genetic distance as a proxy for cultural distance because genetic distances alone may simply reflect the isolation-by-distance effects of spatially-limited gene flow (Wright, 1943) as opposed to shared cultural ancestry.

Third, in the case of migration inflows, we are not convinced that genes are necessarily superior to language for tracking cultural ancestry. The reviewer provides an example of recent immigrants adopting the language of their new nation while retaining many of their other cultural traits, arguing that genes would provide a better marker of culture in this case. However, sociological theories of assimilation suggest that, in some cases, children of immigrants begin to adopt not only the language of the host nation, but also its customs, norms, and values (Alba & Nee, 2003). In these cases, language could at least in principle do a better job than genes of tracking cultural inheritance. Whether this is the case in practice will depend on specifics of the system in question.

Fourth, unlike genes, language is a tangible socially-learned trait. Fundamentally, it is *social learning* that determines the spread of culture (this is what we meant by the “specific underlying process”). Social learning of cultural traits *can* occur intergenerationally alongside

genes, but many cultural traits are also horizontally transmitted through learning from peers. This cultural transmission is not captured by genes.

These are interesting questions to consider and none of this is meant to imply that using genes as a proxy for cultural non-independence is not a useful tool for researchers – on the contrary, some of the most innovative work on this question has made use of genetic rather than linguistic data (Spolaore & Wacziarg, 2009; 2013). We simply want to make the case that controlling for non-independence using linguistic ancestry is a reasonable and principled choice.

These “quibbles” should not distract from the fact you wrote a great paper. Congrats.

We thank this reviewer again for taking the time to read through our manuscript and provide additional feedback. The reviewer’s comments have greatly improved the manuscript.

References

- Alba, R. D., & Nee, V. (2003). *Remaking the American mainstream: Assimilation and contemporary immigration*. Harvard University Press.
- Beck, T., Demirg -Kunt, A., & Levine, R. (2005). SMEs, growth, and poverty: Cross-country evidence. *Journal of Economic Growth*, 10, 199–229.
- Chen, J., Chen, H., Sokal, R. R., Ruhlen, M., & Heaggerty, P. (2012). Worldwide analysis of genetic and linguistic relationships of human populations. *Human Biology*, 84(5), 553–578.
- Gelfand, M. J., Raver, J. L., Nishii, L., Leslie, L. M., Lun, J., Lim, B. C., ... & Yamaguchi, S. (2011). Differences between tight and loose cultures: A 33-nation study. *Science*, 332(6033), 1100–1104.
- Spolaore, E., & Wacziarg, R. (2009). The diffusion of development. *The Quarterly Journal of Economics*, 124(2), 469-529.
- Spolaore, E., & Wacziarg, R. (2013). How deep are the roots of economic development? *Journal of Economic Literature*, 51(2), 325–369.
- Wright, S. (1943). Isolation by distance. *Genetics*, 28(2), 114–138.

REVIEWERS' COMMENTS

Reviewer #3 (Remarks to the Author):

Thank you very much for this great paper! I have nothing more to add.